# T-Cadherin Finetunes Proliferation–Differentiation During Adipogenesis via PI3K–AKT Signaling Pathway

**DOI:** 10.3390/ijms26199646

**Published:** 2025-10-02

**Authors:** Polina Klimovich, Ilya Brodsky, Valentina Dzreyan, Marianna Ivleva, Olga Grigorieva, Mark Meshcheriakov, Ekaterina Semina, Veronika Sysoeva, Vsevolod Tkachuk, Kseniya Rubina

**Affiliations:** 1Faculty of Medicine, Lomonosov Moscow State University, 119991 Moscow, Russiamarianna.ivleva@yandex.ru (M.I.); e-semina@yandex.ru (E.S.); veroniks@mail.ru (V.S.);; 2Institute of Experimental Cardiology, National Medical Research Center of Cardiology Named after Academician E.I. Chazov, 121552 Moscow, Russia; 3Center for Regenerative Medicine, Medical Research and Education Institute, Lomonosov Moscow State University, 119192 Moscow, Russia

**Keywords:** T-cadherin, CDH13, adiponectin, adipogenesis, 3T3-L1, insulin signaling

## Abstract

Adipose tissue renewal requires precise coordination of stem/progenitor cell proliferation, preadipocyte commitment, and terminal adipocyte differentiation. T-cadherin (CDH13), an atypical GPI-anchored cadherin, is expressed in adipose tissue and functions as a receptor for high-molecular-weight (HMW) adiponectin—a key adipokine produced by adipose tissue and involved in metabolic regulation. While T-cadherin is implicated in cardiovascular and metabolic homeostasis, its role in adipogenesis still remains poorly understood. In this study, we used the 3T3-L1 preadipocyte model to investigate the function of T-cadherin in adipocyte differentiation. We analyzed T-cadherin expression dynamics during differentiation and assessed how T-cadherin overexpression or knockdown affects lipid accumulation, expression of adipogenic markers, and key signaling pathways including ERK, PI3K–AKT, AMPK, and mTOR. Our findings demonstrate that T-cadherin acts as a negative regulator of adipogenesis. T-cadherin overexpression ensured a proliferative, undifferentiated cell state, delaying early adipogenic differentiation and suppressing both lipid droplet accumulation and the expression of adipogenic markers. In contrast, T-cadherin downregulation accelerated differentiation, enhanced lipid accumulation, and increased insulin responsiveness, as indicated by PI3K–AKT pathway activation at specific stages of adipogenesis. These results position T-cadherin as a key modulator of adipose tissue plasticity, regulating the balance between progenitor expansion and terminal differentiation, with potential relevance to obesity and metabolic disease.

## 1. Introduction

Adipose tissue plays a central role in systemic energy homeostasis and endocrine regulation. Excess energy is stored in the form of lipids within mature adipocytes of white adipose tissue (WAT). In case of energy deprivation, these lipids are mobilized to supply energy to peripheral tissues. In response to sustained overnutrition, adipose tissue expands to accommodate an increased energy storage, thereby preventing ectopic lipid deposition and lipotoxicity in other organs and tissues. Adipose tissue is not merely passive energy storage but also functions as an endocrine organ, secreting a variety of biologically active molecules, including hormones, adipokines, extracellular matrix proteins, growth factors, and microRNAs [1,2]. Adipokines (adiponectin, leptin, etc.) play critical roles in regulating glucose and lipid metabolism, insulin sensitivity, inflammation, and vascular function. Dysregulated adipokine secretion is closely linked to obesity and its associated comorbidities, such as type 2 diabetes, cardiovascular disease, and metabolic syndrome [3]. Adipose tissue expansion occurs through an increase in adipocyte number (hyperplasia) and/or an increase in adipocyte size (hypertrophy) [4]. Both processes can contribute to adipose tissue expansion, yet hypertrophic growth is more strongly linked to adipose tissue dysfunction, including hypoxia, chronic inflammation, and insulin resistance [3,4,5].

Adipocyte differentiation (adipogenesis) is a multistep process tightly governed by the coordinated transcriptional and signaling networks that control the commitment of progenitor cells to adipocyte lineage, clonal expansion of committed preadipocytes, and terminal differentiation to adipocytes [5]. Insulin is a key endocrine regulator of energy and lipid metabolism, exerting its effects through activating the intracellular signaling cascade that includes the insulin receptor (IR), insulin receptor substrate (IRS) proteins, phosphoinositide 3-kinase (PI3K), and protein kinase B (AKT). In adipose tissue, insulin plays a crucial role in both morphogenesis and function, promoting adipocyte differentiation and supporting the acquisition of a functionally mature adipocyte phenotype [6]. In the context of metabolic syndrome, obesity, and insulin resistance, adipose tissue dysfunction extends beyond the morphological alterations and includes impaired proliferation of stem and progenitor cells, abnormal adipocyte maturation, and disruptions in endocrine function, as reflected by altered secretion profiles of adipokines, growth factors, and microRNAs [2].

Adiponectin is among the most important adipokines abundantly secreted by adipocytes of WAT, exerting insulin-sensitizing, anti-inflammatory, and anti-atherogenic effects [7]. Circulating adiponectin levels are typically high in healthy individuals but decline significantly in conditions such as obesity, type 2 diabetes, and cardiovascular disease. Low adiponectin levels are strongly associated with insulin resistance and an increased risk of metabolic complications [5]. Adiponectin exists in multiple forms including trimeric (low molecular weight), hexameric (medium), and high molecular weight (HMW), the latter being considered the most metabolically active form [5,7,8,9]. Adiponectin exerts its effects through two “classical” receptors, AdipoR1 and AdipoR2, which activate the key downstream signaling pathways, including AMP-activated protein kinase (AMPK) and peroxisome proliferator-activated receptor-α (PPAR-α) [8,10]. Importantly, T-cadherin has been identified as a specific receptor for high-molecular-weight (HMW) adiponectin [11], although the downstream signaling pathways triggered by this interaction remain largely unexplored.

T-cadherin (also known as cadherin-13 or H-cadherin) is an atypical member of the cadherin superfamily. Unlike “classical cadherins”, which are transmembrane glycoproteins, T-cadherin lacks both transmembrane and intracellular domains and is tethered to the plasma membrane via a glycosylphosphatidylinositol (GPI) anchor [5]. Due to this unique structure, T-cadherin is not involved in classic cell–cell adhesion but is thought to operate as a signaling receptor, which sequesters the circulating adiponectin on the cell surface of organ and tissues, thereby facilitating its protective effects [11,12,13]. Supporting this, a substantial body of genetic and experimental evidence underscores the important role of T-cadherin in metabolic health. Genome-wide association studies revealed that reduced systemic HMW adiponectin was associated with single nucleotide polymorphisms (SNPs) in CDH13 gene, potentially suggesting the existence of an autocrine signaling loop [5,12,13,14].

In fact, T-cadherin binds two structurally and functionally distinct ligands: HMW adiponectin and low-density lipoprotein (LDL). This dual ligand specificity suggests that T-cadherin may act as a molecular “switch,” modulating cellular responses based on the plasma concentrations of circulating ligands. Under physiological conditions, where adiponectin levels are high—as in healthy individuals—T-cadherin mediates cardiovascular and metabolic protection. In contrast, under pathological conditions such as obesity or metabolic syndrome with elevated LDL in the bloodstream, T-cadherin may drive pathological changes [5]. Upon binding LDL, T-cadherin initiates intracellular signaling cascades, triggering calcium influx or ERK1/2 phosphorylation or NF-κB activation that collectively stimulate cell proliferation with potentially pro-atherosclerotic effects in vessel walls [15,16].

Although the roles of T-cadherin in the cardiovascular and nervous systems are well established [5,17], its function in adipose tissue, particularly in the light of progenitor/stem cell renewal and adipocyte differentiation, remains poorly elucidated [5]. Single-cell RNA sequencing data reported T-cadherin expression in mesenchymal stem cells (MSCs) derived from various sources including adipose tissue [18,19,20]. In our recent study, we demonstrated that adipose-derived murine MSCs, lacking the full-size T-cadherin, are more inclined to differentiate along the adipogenic lineage than control cells, a tendency marked by greater lipid-droplet accumulation and altered levels of the key early and late adipogenic markers [2]. We also explored the effects of T-cadherin ligands (LDL and adiponectin) on adipogenic differentiation and demonstrated the existence of a feedback loop mediated by T-cadherin, supporting the idea that T-cadherin acts as a key mediator linking extracellular metabolic cues to adipocyte differentiation [2]. Additional evidence indicates that altered T-cadherin levels in adipose tissue and circulation can reflect lipid-metabolism disturbances and are linked to metabolic disorders [21,22]. For example, Göddeke et al. demonstrated that T-cadherin mRNA levels are reduced in the visceral adipose tissue of both obese mice and humans [21]. In the same study, siRNA-mediated knockdown of T-cadherin in 3T3-L1 preadipocytes prior to differentiation led to a decreased expression of key adipogenic transcription factors, PPARγ and C/EBPα, and significantly impaired the terminal adipocyte differentiation. Additionally, T-cadherin was shown to affect fatty acid uptake and lipid accumulation during differentiation [21], highlighting the functional role of T-cadherin in metabolic remodeling during adipogenesis and regulation of adipose tissue homeostasis.

The present study aims to examine the expression dynamics and functional role of T-cadherin during adipogenesis using the well-established 3T3-L1 mouse preadipocyte model. Specifically, we analyzed: (1) temporal changes in T-cadherin expression across distinct stages of differentiation, and evaluated its impact on (2) lipid droplet formation, (3) the expression of early and late adipogenic markers, and (4) the activation of key signaling pathways involved in adipocyte differentiation and maturation, including downstream components of the insulin/IGF-1 axis such as APPL1, PI3K, AMPK, ERK, and AKT. Elucidating the role of T-cadherin in adipogenesis may uncover novel regulatory mechanisms governing adipose tissue plasticity and provide a valuable insight into how metabolic cues are integrated during adipocyte renewal, differentiation, and function. Ultimately, these findings could contribute to identifying the new therapeutic targets for obesity and clarify key aspects of metabolic dysfunction.

## 2. Results

### 2.1. T-Cadherin Expression Increases upon Induction of Adipocyte Differentiation

To explore the role of T-cadherin, we first evaluated its expression during the differentiation of wild-type (WT) 3T3-L1 cells, a well-established murine pre-adipocyte cell line widely used to model adipogenesis [23]. Differentiation was induced by an adipogenic cocktail as described in Materials and Methods, and cell lysates were collected on days 3, 7, 10, and 14. Western blots revealed that T-cadherin began to rise on day 7 in adipogenic medium (a/m), peaking on day 14 (Figure 1). Densitometry showed a 1.5-fold increase on day 7 (*p* = 0.0036 vs. day 0; *p* = 0.0361 vs. day 3), a 2.5-fold increase on day 10 (*p* = 0.0015 vs. day 0; *p* = 0.0085 vs. day 3), and a 3.5-fold increase on day 14 (*p* = 0.0019 vs. day 0; *p* = 0.0126 vs. day 3) compared to day 0 (Figure 1). In contrast, T-cadherin levels remained stable in cell cultures maintained in control medium (c/m) and were significantly lower compared to the corresponding cells cultured in adipogenic medium (a/m) (* *p* < 0.05). These data suggest that T-cadherin may play different roles at the early versus later stages of adipogenic differentiation.

### 2.2. Silencing T-Cadherin Accelerates Lipid-Droplet Accumulation During Adipogenic Differentiation

Adipogenic differentiation in 3T3-L1 model is typically evaluated by (i) lipid droplet accumulation, (ii) expression of early and late adipogenic markers, and (iii) activation of insulin signaling pathways such as the Ras/ERK1/2 and phosphoinositide-3-kinase (PI3K)/Akt [24,25,26].

To define the contribution of T-cadherin to adipogenesis, we first obtained pooled transfectant populations with different levels of T-cadherin (T-cadherin overexpressing cells—Tcad↑, and cells with suppressed T-cadherin—SH-Tcad↓) as discussed in Materials and Methods section. To evaluate the levels of T-cadherin we performed Western blot analysis and confirmed that the plasmid transfection effectively altered T-cadherin expression (Figure 2).

Next, we induced adipogenic differentiation in the obtained 3T3-L1 pooled transfectant populations (Tcad↑ and SH-Tcad↓), control cells transfected with a control plasmid, and WT cells. We stained these cells for neutral lipids with Oil Red O on day 7. The Oil Red O staining revealed marked differences in lipid droplet accumulation (Figure 3). While WT cells contained many uniformly distributed small-to-medium lipid droplets, SH-Tcad↓ cells, in contrast, displayed markedly larger droplets as early as day 7. Although a subset of Tcad↑ cells accumulated lipid droplets, these droplets gathered in discrete clusters, and many cells in the same field remained undifferentiated.

The heterogeneity in droplet size and morphology prevented an accurate quantification of lipid-positive cells. Both manual counting (lipid-positive cells and normalizing to the total cell number) and neural-network analysis proved unreliable [2]; therefore, no numerical data are provided. Nonetheless, a clear trend emerged: silencing T-cadherin increased lipid droplet accumulation, whereas its overexpression reduced the fraction of cells undergoing adipogenic differentiation. Transfection with a control vector (cont) had no impact on lipid formation. Collectively, these findings suggest that T-cadherin may act as a negative regulator of adipogenesis.

### 2.3. T-Cadherin Overexpression Enhances Proliferative Capacity of 3T3-L1 Cells

Although proliferation expands the pool of progenitors available for adipogenesis, intensive mitotic activity can hinder differentiation: the rapidly dividing cells are less prone to be committed towards adipocyte lineage [27,28]. To determine how T-cadherin levels affect 3T3-L1 proliferation, we monitored real-time impedance (Cell Index) with the xCELLigence system for 98 h using pooled transfectant populations with different T-cadherin levels (Tcad↑, SH-Tcad↓, WT and cont cells). Cells were seeded at the same density of 5 × 10^3^ cells per well.

During the initial 0–3 h attachment/spreading phase, Tcad↑ cells showed a higher spreading index than any other group, whereas no difference was detected among WT, control (cont), and SH-Tcad↓ cells (Figure 4). In the main proliferative phase (10–30 h), impedance analysis revealed that Tcad↑ cells divided more rapidly than WT, SH-Tcad↓ and control cells (cont), while WT, cont, and SH-Tcad↓ populations displayed comparable growth. The difference in proliferative rate between Tcad↑ and SH-Tcad↓ cells may reflect the accelerated adipogenesis in the cells with suppressed T-cadherin and increased lipid-droplet accumulation, while the opposite trend was observed in the cells overexpressing T-cadherin.

### 2.4. Generation of 3T3-L1 Clones with Altered T-Cadherin Expression

Since standard plasmid transfection typically yields only about 10% of cells expressing the construct, we next generated 3T3-L1 cell clones using the pooled transfectant popshowed the strongest change in T-cadherin expression. The final panel comprised two overexpressing cell lines (G7-Tcad↑ and G9-Tcad↑), two clones with suppressed T-cadherin (SH1-Tcad↓ and SH4-Tcad↓), and control clone. Relative to WT and cont cells, T-cadherin mRNA increased 3.5-fold in G7-Tcad↑ and 1.9-fold in G9-Tcad↑, while in SH1-Tcad↓ and SH4-Tcad↓ T-cadherin mRNA decreased 3.3- and 4.0-fold, respectively. The control vector had no effect on T-cadherin protein or mRNA levels (Figure 5).

### 2.5. T-Cadherin Overexpression Delays Adipogenic Differentiation in 3T3-L1 Cells

To further explore the role of T-cadherin in adipogenic differentiation we quantified the mRNA levels of key adipogenic markers in the generated clones. The early marker pparγ and the late markers adipoQ (encodes adiponectin), plin-1 (encodes perillipin-1), and lep (encodes leptin) were measured on days 0, 3, 7, and 10 after adipogenic induction (Figure 6). Pparγ is widely recognized as a master regulator of adipogenesis, essential for driving pre-adipocytes through their conversion into mature adipocytes [29,30].

Pparγ mRNA transcripts increased progressively in all samples as adipogenic differentiation proceeded (Figure 6A). Throughout the time course, WT, cont, and SH4-Tcad↓ cells maintained similar pparγ expression levels, whereas the overexpressing clones G7-Tcad↑ and G9-Tcad↑ consistently displayed significantly less pparγ mRNA starting at baseline (day 0) (4.2 and 3.0-fold lower) and continuing through days 3 (2.7- and 3.0-fold lower), and 7 (2.0- and 3.5-fold lower, correspondingly), and 10 (2.5- and 4.8-fold lower, correspondingly) compared to WT cells (*p* < 0.05; Figure 6A). Of note, cells maintained in control medium showed no change in pparγ expression levels.

Adiponectin and leptin are canonical markers of functionally mature adipocytes [31,32], therefore we next quantified their mRNA content in 3T3-L1 clones with different levels of T-cadherin expression on days 0, 3, 7, and 10 after induction of adipogenic differentiation.

AdipoQ transcripts appeared by day 3 after adipogenic induction in WT, cont, and SH4-Tcad↓ cells (Figure 6B). By day 3, mRNA expression levels were already 4.1-fold higher in SH4-Tcad↓ than in WT cells (*p* < 0.05). In T-cadherin overexpressing clones G7-Tcad↑ and G9-Tcad↑, adipoQ mRNA emerged later, on day 7, and was 13.7- and 6.9-fold lower, correspondingly, compared to WT cells (*p* < 0.05). This tendency persisted through day 10: adipoQ mRNA remained significantly elevated in SH4-Tcad↓ clone and was consistently suppressed in G7-Tcad↑ and G9-Tcad↑, relative to both WT and control cells (*p* < 0.05) (Figure 6B). Of note, adipoQ mRNA was detected only in the cells cultured in adipogenic medium.

A similar pattern was noted for lep mRNA. Lep transcript levels rose by day 3 and continued to increase through day 10 in all cell types, yet it remained 2- to 7-fold lower (*p* < 0.05) in T-cadherin overexpressing clones (G7-Tcad↑, G9-Tcad↑) compared to WT cells (Figure 6C). In SH4-Tcad↓ cells, lep mRNA on day 10 was approximately 1.7 times higher than in WT cells. By day 10, the lep mRNA levels in G7-Tcad↑, and G9-Tcad↑ clones were 2.1- and 2.8-fold lower than in WT cells.

Perilipin, also referred to as lipid-droplet–associated protein, is encoded in mice by the plin gene (https://www.ncbi.nlm.nih.gov/gene/103968 accessed on the 2 June 2025). Members of the perilipin family line the surface of lipid droplets, and their phosphorylation is critical for lipid droplet formation [33]. Interestingly, plin-1 mRNA level began to rise by day 3 in all groups in adipogenic conditions: the transcript levels were already higher in both T-cadherin overexpressing clones (G7-Tcad↑, G9-Tcad↑) (3.6- and 3.9-fold, correspondingly) and in SH4-Tcad↓ clone (3.4-fold) compared to WT cells (*p* < 0.05) (Figure 6D). The increase persisted up to day 7, with plin-1 mRNA remaining markedly elevated in G9-Tcad↑ (2.9- fold) and in SH4-Tcad↓ (2.1-fold) versus WT (*p* < 0.05). By day 10, plin-1 expression in G7-Tcad↑ and G9-Tcad↑ returned to WT levels, whereas it remained 2.2-fold higher in SH4-Tcad↓ clone (*p* < 0.05) (Figure 5). Of note, in cells maintained in control medium, plin-1 and lep transcripts were barely detectable and did not vary over the 10-day period. 

Therefore, T-cadherin suppression accelerates 3T3-L1 adipogenic differentiation and maturation: adipoQ, lep, and plin-1 transcripts appear sooner and remain higher throughout differentiation. Conversely, T-cadherin overexpression delays and diminishes expression of these adipogenic markers thus slowing adipocyte maturation. Overall, these data suggest that T-cadherin may act as a negative regulator of adipogenic differentiation.

### 2.6. T-Cadherin Suppression Pre-Activates the Insulin-Signaling Cascade

Having shown that T-cadherin modulates adipogenic differentiation in 3T3-L1 pre-adipocytes, we further explored the activation of key signaling pathways that govern this process. Components downstream of the insulin/IGF-1 pathway, such as APPL1, PI3K, AMPK, ERK and AKT are essential for adipogenesis [34]. We next studied the activation of these signaling pathways (APPL1, PI3K, total and phosphorylated forms of AMPK, ERK, and AKT) in lysates from cells with different levels of T-cadherin cultured either in control medium or in adipogenic conditions. Control vector clone was further omitted from the representative Western blots since no effects of control plasmid transfection were noticed in prior experiments. Clones G7-Tcad↑, G9-Tcad↑, SH4-Tcad↓ and WT cells were induced towards adipogenic differentiation, and cell lysates were collected on days 0, 3, 7, and 14 (Figure 7, Figure 8 and Figure 9). Under basal conditions (day 0), cells with suppressed T-cadherin (SH4-Tcad↓) displayed strong kinase activation compared to WT cells. Specifically, ERK1/2 was activated and the level of pERK1/2 was elevated 3.3-fold (1.639 ± 0.261 vs. 0.478 ± 0.022; * *p* = 0.047) compared to WT (Figure 7A,B). Similarly, pAKT was increased 4-fold (2.408 ± 0.258 vs. 0.592 ± 0.310; * *p* = 0.046) (Figure 7A,C), and pAMPK was upregulated almost 7-fold (3.305 ± 0.505 vs. 0.468 ± 0.132; * *p* = 0.032) (Figure 7A,D). By contrast, APPL1 (an adaptor linking adiponectin and insulin signaling [35]), was markedly decreased (0.291 ± 0.253 vs. 1.424 ± 0.032; * *p* = 0.047) in SH4-Tcad↓ compared to WT (Figure 7A,E). Given that ERK1/2 and AKT activation is required for the early cell-cycle re-entry during adipogenesis [27,36], these data on the elevated pERK1/2 and pAKT suggest that downregulating T-cadherin primes cells for adipogenic differentiation. The clones overexpressing T-cadherin (G9-Tcad↑ and G7-Tcad↑) did not reveal comparable changes.

PI3K, the principal downstream effector of insulin receptor [37], was significantly reduced only in G9-Tcad↑ cells compared to WT cells (0.186 ± 0.113 vs. 1.307 ± 0.234; * *p* = 0.050). No differences were detected in SH4-Tcad↓ or G7-Tcad↑ clones compared to WT cells (Figure 7A,F).

By day 7 of adipogenic induction, APPL1 levels had fallen 1.7-fold in WT cells (1.094 ± 0.087 vs. 1.884 ± 0.092; * *p* = 0.0246) and 2.4-fold in SH4-Tcad↓ cells (0.349 ± 0.102 vs. 0.843 ± 0.052; * *p* = 0.0494), while remaining unchanged in G9-Tcad↑ and G7-Tcad↑ relative to the same cell types in control medium (Figure 8A,C). Of note, under the same adipogenic conditions, APPL1 in SH4-Tcad↓ was three times lower than in WT cells (0.349 ± 0.102 vs. 1.094 ± 0.087; ** *p* = 0.0078), consistent with our earlier assumption that cells with suppressed T-cadherin are more predisposed to adipogenic differentiation and aligning with the previous reports that APPL1 declines during adipocyte differentiation [38].

Moreover, on day 7, phospho-AKT levels decreased in WT cells cultured in adipogenic medium compared with WT cells in control medium (0.498 ± 0.029 vs. 0.936 ± 0.003; ** *p* = 0.00447). The reduction was not statistically significant in G9-Tcad↑ and G7-Tcad↑ or SH4-Tcad↓ cells, indicating no clear dependence on T-cadherin expression (Figure 8A,D).

Furthermore, PI3K content was unchanged in G9-Tcad↑, G7-Tcad↑ and WT cells cultured in control medium compared to adipogenic conditions (Figure 8A,B). However, in SH4-Tcad↓ PI3K level rose 12-fold upon induction of adipogenic differentiation compared with control medium (0.694 ± 0.146 vs. 0.055 ± 0.016; * *p* = 0.0487) and was likewise 9-fold higher than in WT cells (0.694 ± 0.146 vs. 0.079 ± 0.029; ** *p* = 0.005).

Phosphorylated AMPK and ERK were barely detectable on day 7 (Figure 8A), and the total AMPK and ERK levels remained unchanged throughout differentiation, irrespective of T-cadherin expression (Figure 8A).

Under adipogenic conditions on day 14, PI3K levels decreased in G9-Tcad↑ (0.40 ± 0.10 vs. 1.20 ± 0.01; * *p* = 0.046), G7-Tcad↑ (0.40 ± 0.10 vs. 1.26 ± 0.04; * *p* = 0.016), and WT cells (0.417 ± 0.053 vs. 0.890 ± 0.010; * *p* = 0.013) relative to the same cell types in control medium (Figure 9A,B). By contrast, SH4-Tcad↓ cells displayed the opposite response: PI3K rose significantly in adipogenic conditions compared with control medium (1.522 ± 0.178 vs. 0.50 ± 0.10; * *p* = 0.038) and was three-fold higher than in WT in adipogenic conditions (1.522 ± 0.178 vs. 0.417 ± 0.053; *** *p* = 0.0004) (Figure 9A,B).

Concurrently, by day 14 AKT phosphorylation increased in every cell line under adipogenic conditions, whereas only trace AKT phosphorylation was detected in control medium (Figure 9A,D). Notably, phospho-AKT in SH4-Tcad↓ was significantly higher than in WT cells, both in adipogenic conditions (0.905 ± 0.005 vs. 0.512 ± 0.012; * *p* = 0.0211). By day 14, APPL1 levels remained unchanged, irrespective of the medium or T-cadherin expression (Figure 9A,C).

### 2.7. T-Cadherin Expression Modulates mTOR Signaling Pathway in 3T3-L1 Cells with Different Levels of T-Cadherin Expression

It is well-established that the PI3K–mTOR axis orchestrates cell differentiation, proliferation, and metabolism in response to extracellular cues [39]. In light of our findings demonstrating that the altered T-cadherin expression impacts 3T3-L1 cell proliferation and adipogenic differentiation, followed by changes in PI3K activation during adipogenesis, we next explored mTOR mRNA content as a downstream effector of this pathway. Given that mTOR is positioned immediately downstream of PI3K in the insulin/IGF-1 pathway [39], we therefore assessed mRNA mTOR basal levels and its downstream partners Rptor and Rictor in 3T3-L1 cells with different levels of T-cadherin expression.

T-cadherin overexpression elevated mTOR mRNA levels (significantly in G9-Tcad↑ and modestly in G7-Tcad↑), whereas T-cadherin suppression downregulated mTOR mRNA in SH4-Tcad↓ (Figure 10A). Conversely, Rptor and Rictor transcripts declined in G9-Tcad↑ and G7-Tcad↑ but remained unchanged in SH4-Tcad↓ (Figure 10B,C). These findings imply that T-cadherin may influence mTOR signaling indirectly via the PI3K–mTOR axis, thereby modulating cellular insulin sensitivity [40].

## 3. Discussion

A growing body of evidence indicates that differentiation of preadipocytes into mature fat cells is a complex process involving an interplay of regulatory factors that exert both positive and negative effects on a network of signaling pathways converging on the adipogenic gene program. Upon exposure to adipogenic inducers, 3T3-L1 cells proceed through three key stages: commitment to the preadipocyte lineage, clonal expansion of the committed cells, and terminal differentiation marked by the expression of adipocyte-specific genes and intracellular triglyceride accumulation [36,40]. Our findings demonstrate that T-cadherin expression is dynamically regulated during adipogenesis and T-cadherin exerts a stage-specific effects on 3T3-L1 differentiation. T-cadherin levels rose sharply in mid-to-late differentiation of 3T3-L1 WT cells (days 7–10), suggesting its functional role in the transition from proliferating preadipocytes to mature adipocytes (Figure 1). This elevated expression suggests that T-cadherin may coordinate the late stages of adipogenesis by regulating cell–cell or cell–matrix interactions as differentiating cells round up and accumulate lipids. Of note, a modest increase in T-cadherin expression on day 7 in control medium may reflect its role in contact inhibition—particularly relevant under our experimental conditions, where cells were seeded and maintained in culture for 14 days without passaging. This explanation is supported by the earlier findings demonstrating that T-cadherin mediates contact inhibition of endothelial cell migration during angiogenesis through interactions with the surrounding stromal cells [41,42,43].

We further examined the role of T-cadherin in adipogenic differentiation by experimentally altering T-cadherin levels. We found that T-cadherin silencing accelerated lipid-droplet formation (Figure 3) and adipogenic marker expression (Figure 6), whereas its overexpression delayed these processes. Preadipocytes with suppressed T-cadherin accumulated conspicuously larger lipid droplets as early as day 7 (Figure 3), and showed earlier and higher expression levels of adiponectin, leptin, and perilipin mRNAs (Figure 6), consistent with the accelerated adipocyte differentiation. In comparison, T-cadherin overexpression resulted in fewer cells with lipid droplets (Figure 3) and significantly downregulated adipogenic markers (pparγ, adipoQ, lep, plin-1) (Figure 6), indicating a differentiation delay [34]. Of note, control vector-transfected cells demonstrated the same levels of adipogenic markers as WT cells, confirming that these effects were T-cadherin-specific.

In contrast to our findings, Gödekke et al. [21] reported that transient siRNA-mediated knockdown of T-cadherin in 3T3-L1 cells reduced pparγ and c/ebpα expression, thereby impairing terminal differentiation. This apparent discrepancy likely reflects methodological differences. Gödekke et al. employed transient siRNA knockdown either immediately before adipogenic induction (day 0) or during the early induction phase (day 4). Such short-term T-cadherin suppression at the early stages of adipogenesis may affect the expression of early adipogenic factors, thereby impairing initiation of the adipogenic program. By contrast, we established stable shRNA-mediated knockdown clones through antibiotic selection and single-cell cloning. The clones maintained suppressed T-cadherin expression throughout both the expansion and differentiation stages, providing a consistent background for long-term analysis. Furthermore, the adipogenic induction protocols differed substantially: Gödekke et al. applied a cocktail containing troglitazone, hydrocortisone, transferrin, T3, and insulin, whereas we utilized the classical, widely used differentiation mixture of IBMX, dexamethasone, insulin, and FBS [30].

Our findings are consistent with our previously published work on the role of T-cadherin in the adipogenic differentiation of mesenchymal stem/stromal cells (MSCs) [2]. Specifically, we have demonstrated that T-cadherin deficiency facilitated adipogenic differentiation of MSCs, as evidenced by elevated adiponectin and leptin secretion, along with characteristic morphological changes. Therefore, our combined results—both previous and present—support the notion that the cells lacking full-size T-cadherin are predisposed to adipogenic lineage commitment. One possible explanation is that the inhibitory effects of T-cadherin overexpression may be facilitated by signals arising from T-cadherin–mediated homophilic adhesion, which serve to maintain preadipocytes in an undifferentiated state. Such adhesion-dependent signaling could sustain anti-adipogenic pathways or preserve a progenitor-like transcriptional profile. Of note, cell adhesion molecules are well-known regulators of adipocyte commitment through their influence on Wnt/β-catenin signaling and other pathways that actively suppress adipogenesis [44]. We previously demonstrated that T-cadherin regulates endothelial monolayer permeability. Overexpression of T-cadherin induced phosphorylation of VE-cadherin at Y731—a key site for β-catenin binding—thereby activating Rho GTPases signaling, actin cytoskeleton remodeling and VE-cadherin endocytosis, ultimately resulting in disruption of the endothelial barrier [45]. In this context, the endogenously elevated T-cadherin expression during differentiation may function as a feedback mechanism through cell–cell adhesion signals that restricts an excessive or premature adipogenic conversion in vitro and in vivo. Collectively, our results implicate T-cadherin as a suppressor of adipogenic program: its downregulation promotes commitment to the adipocyte lineage, whereas its overexpression impedes adipogenesis.

Next, we demonstrated that T-cadherin is involved in the interplay between differentiation and proliferation in 3T3-L1 cells. Although cell proliferation and adipogenic differentiation are generally considered mutually exclusive processes, a functional cross-talk between them has been well established [27]. In adipogenic conditions, prior to terminal differentiation growth-arrested preadipocytes re-enter the cell cycle and undergo several rounds of division (clonal expansion) before committing to the adipogenic program. Our results highlight a dual role for T-cadherin in both the differentiation process itself and in the proliferative phase preceding differentiation. T-cadherin overexpression significantly enhanced 3T3-L1 proliferation, as evidenced by increased impedance-based cell index during both the initial attachment/spreading stage and the main proliferative phase (10–30 h) (Figure 4). In contrast, T-cadherin-deficient cells, WT, and vector control (cont) cells exhibited comparable proliferation rates, suggesting that endogenous T-cadherin is not a limiting factor for proliferation under basal conditions. While T-cadherin deficiency does not significantly affect cell proliferation, T-cadherin overexpression appears to stimulate cell cycle progression. These findings are consistent with the previous studies showing that T-cadherin overexpression in endothelial cells, vascular smooth muscle cells, and pericytes enhanced proliferation, migration, and survival via activation of pro-mitogenic pathways or modulation of key signaling checkpoints involved in cell cycle control [46,47,48,49].

Literature provides abundant data underscoring the importance of the precise timing for proliferation–differentiation transition during adipogenesis. In 3T3-L1 cells, transient activation of the ERK/MAPK signaling pathway within the first ~12 h of adipogenic induction is essential for mitotic clonal expansion of preadipocytes and upregulation of key adipogenic transcription factors, including C/EBPβ/δ and PPARγ [50]. However, if ERK/MAPK mitogenic signaling persists beyond this early stage, it can lead to the phosphorylation and inactivation of C/EBPs, PPARγ, and other differentiation-related factors, thereby impairing terminal adipocyte differentiation [50,51]. Our data indicate that, at baseline prior to adipogenic stimulation (day 0), T-cadherin-deficient cells (SH4 Tcad↓) exhibited elevated ERK1/2 phosphorylation (compared to WT and T-cadherin overexpressing cells), presumably priming them for the early clonal expansion and differentiation (Figure 7). In contrast, G7-Tcad↑ and G9-Tcad↑ cells maintained sustained ERK activation during mitotic phase and later (day 7), consistent with their increased proliferation and attenuated differentiation (Figure 7 and Figure 8). These data suggest that prolonged ERK signaling in T cadherin-overexpressing cells may reinforce the proliferative state, whereas its early decline in SH4 Tcad↓ facilitates differentiation (day 7). In the context of adipose tissue homeostasis, these effects of T-cadherin may help to set the balance between progenitor expansion and differentiation commitment—a dynamic interplay that is central to adipose tissue plasticity.

We further explored the role of T-cadherin in regulating key adipogenic signaling pathways. The adipogenic effects of insulin are primarily mediated through PI3K/AKT signaling, which upregulates PPARγ and C/EBPα—the main transcription factors governing terminal adipocyte differentiation and the activation of mTORC1 [1]. The PI3K-AKT axis plays a central role in mediating insulin effects on both metabolism and cell proliferation, and its activation is considered both necessary and sufficient to induce adipocyte differentiation in vitro [1,37]. We examined PI3K and AKT expression and activation levels as critical nodes within the insulin signaling cascade. At baseline (day 0), T-cadherin-deficient cells exhibited a marked pre-activation of the insulin signaling pathway. Even in the absence of adipogenic inducers (day 0), AKT phosphorylation was markedly elevated in SH4-Tcad↓ cells, significantly exceeding the levels detected in WT and T-cadherin-overexpressing cells (G7-Tcad↑ and G9-Tcad↑) (Figure 7). However, upon adipogenic induction (day 7), SH4-Tcad↓ cells did not show a statistically significant increase in pAKT compared to WT cells. We attribute this to the baseline differences (day 0): even prior to induction, SH4-Tcad↓ cells exhibited elevated pAKT levels relative to WT (Figure 8). In contrast, upon adipogenic induction (day 7), the level of PI3K rose significantly in T-cadherin deficient cells compared to WT or T-cadherin overexpressing cells, most likely reflecting their enhanced response to the insulin present in the induction cocktail (Figure 8). By day 14, (the end point of the differentiation timeline) cumulative effect of T-cadherin knockdown on the PI3K–AKT pathway became clearly evident. Both PI3K protein expression and phospho-AKT levels were significantly higher in SH4-Tcad↓ adipocytes compared to WT (Figure 9). This activation pattern indicates that, in the absence of T-cadherin, cells exhibited an elevated insulin-dependent activation of the PI3K/AKT signaling or lowered threshold for activation.

Our data therefore suggest that T-cadherin suppression “primes” cells for adipogenic differentiation by elevating AKT activity to the levels normally seen only after induction, while the elevated PI3K indicates that T-cadherin-deficient cells exhibit a more robust commitment to adipogenesis. In T-cadherin overexpressing cells, PI3K levels were modestly and stage-specifically altered rather than consistently decreased, suggesting that elevated T-cadherin shifts the signaling equilibrium without fully suppressing PI3K–AKT activity (Figure 7, Figure 8 and Figure 9). The overall trend of decreased PI3K/AKT signaling in G7-Tcad↑ and G9-Tcad↑ cells, correlating with their reduced adipogenic potential (Figure 7, Figure 8 and Figure 9), suggests that T-cadherin elevated expression may impose a restraint on the insulin/PI3K/AKT axis, both prior to and during adipogenesis.

Our data are in line with the previous studies, where T-cadherin overexpression and homophilic interaction in human endothelial cells attenuated insulin-dependent activation of the PI3K/Akt/mTOR signaling axis, along with the reduced eNOS production, cell migration, and angiogenesis in vitro. In contrast, T-cadherin silencing enhanced insulin sensitivity, suggesting that the reduced responsiveness to insulin in T-cadherin-overexpressing cells may result from a chronic activation of the Akt/mTOR-dependent negative feedback loop within the insulin signaling cascade [52]. Additionally, co-immunoprecipitation assay revealed a physical association between T-cadherin and the insulin receptor, with both localized to lipid raft domains of the plasma membrane in vascular endothelial cells [52]. Our findings suggest that T-cadherin could serve as a modulator of PI3K-AKT signaling pathway. Presuming that T-cadherin expression inversely correlates with PI3K-AKT activation, elevated T-cadherin levels may locally impair insulin sensitivity in preadipocytes, whereas downregulated T-cadherin may enhance it. This relationship is particularly significant given that the insulin-PI3K-AKT signaling cascade regulates glucose uptake and lipogenesis, and its impairment is a hallmark of insulin resistance [53]. Future studies are warranted to investigate whether T-cadherin levels in adipose tissue correlate with systemic insulin sensitivity or adiponectin responsiveness, especially in light of T-cadherin being a receptor for HMW adiponectin, the most metabolically active form of adiponectin [5,53,54].

Towards that end, a similar pre-activation pattern was observed for AMPK in T-cadherin deficient cells: at baseline (day 0), phospho-AMPK was an order of magnitude higher in SH4-Tcad↓ cells compared to WT cells, G7-Tcad↑, and G9-Tcad↑ (Figure 7). At the first glance, this may appear paradoxical, as AMPK is known to suppress adipogenesis by inhibiting the early mitotic clonal expansion phase, leading to reduced expression of both early (C/EBPs and PPARγ) and late adipogenic markers (fatty acid synthase (FAS), sterol regulatory element-binding protein-1c (SREBP-1c), and the intracellular lipid chaperone aP2) [4,55]. However, at later stages of differentiation, this early elevation in AMPK activity did not impair adipogenesis (Figure 3, Figure 6, Figure 8 and Figure 9). A plausible explanation is that both the timing and cellular context of AMPK activation are critical. In T-cadherin deficient cells, AMPK was pre-activated at confluence in control medium conditions (day 0), but this activation diminished rapidly following the exposure to adipogenic medium comprising insulin (Figure 7). Supporting this assumption, by days 7 and 14, phospho-AMPK levels were negligible across all cell lines, suggesting that the initial AMPK activation in T-cadherin deficient cells was not sustained throughout differentiation. Recent studies have also shown that AMPK plays a pivotal role in maintaining mitochondrial homeostasis and insulin sensitivity [56,57]. Therefore, the elevated phospho-AMPK level detected at baseline in SH4-Tcad↓ cells is more likely indicative of AMPK’s role in supporting metabolic functions, such as mitochondrial biogenesis, or insulin responsiveness rather than adipogenic differentiation, particularly given that phospho-AMPK levels declined following the induction of adipogenesis (days 7, 14).

Additionally, T-cadherin deficiency resulted in a reduced level of APPL1 expression both at baseline (Figure 7) and on day 7 following adipogenic induction (Figure 8). By contrast, APPL1 levels remained consistently elevated in T-cadherin-overexpressing cells at day 7, with no evident decline (Figure 8). APPL1 is an adaptor protein that directly interacts with adiponectin receptors (AdipoR1 and AdipoR2) and plays an important role in intracellular signaling. Upon adiponectin stimulation, the COOH-terminal domain of AdipoR1 interacts with adiponectin, while its intracellular NH_2_-terminal domain binds APPL1. APPL1 binds not only to AdipoR1 and AdipoR2 but also to insulin receptor. This interaction is well-characterized and enables APPL1 to mediate multiple signaling cascades, including activation of AKT, PI3K, insulin receptor substrates (IRS1/2), AMPK, p38 MAPK, and GLUT4 membrane translocation [35,58]. This crosstalk between insulin signaling, mediated by the PI3K-AKT pathway, and adiponectin signaling, mediated by the APPL1-AMPK axis, constitutes a key mechanism by which adiponectin enhances insulin sensitivity in the target tissues [35,58,59]. Several studies have addressed the role of APPL1 in regulating adipogenic differentiation, yielding both consistent and context-dependent findings. Research by Wen et al. and Lin et al. demonstrated that adipogenic differentiation capacity of MSCs and 3T3-L1 preadipocytes was diminished following APPL1 knockdown [38,60]. Conversely, other studies showed that APPL1 knockdown enhanced adipogenesis. For example, adiponectin was demonstrated to enhance osteogenic differentiation in human adipose-derived stem cells by activating the APPL1-AMPK signaling pathway [61]. In line with these findings, APPL1 knockdown was reported to promote adipogenesis in cultured human bone marrow-derived MSCs by inhibiting autophagy flux and disrupting the balance between adipogenic and osteogenic differentiation—an effect of APPL1 implicated in the pathogenesis of osteoporosis [38]. Our findings support a link between the enhanced adipogenic potential of T-cadherin deficient cells (earlier and increased lipid accumulation) and their diminished APPL1 expression, whereas elevated APPL1 levels in T-cadherin overexpressing cells were associated with reduced adipogenic capacity and increased proliferation. Given that APPL1 functions as an important cytoplasmic adaptor linking adiponectin receptors (AdipoRs) and the insulin receptor, further investigation is needed to elucidate the molecular mechanisms through which T-cadherin modulates this signaling axis. As a GPI-anchored protein, T-cadherin localizes to lipid raft domains, where it may recruit or immobilize signaling complexes, including APPL1, thereby limiting the availability for downstream signaling [5]. Exploring whether APPL1, AdipoRs, PI3K or insulin receptor colocalize with T-cadherin within the plasma membrane may provide valuable mechanistic insights into how T-cadherin affects these signaling pathways.

Given the upstream changes in the PI3K–AKT signaling axis in cells with different levels of T-cadherin expression (Figure 7, Figure 8 and Figure 9), we next analyzed the mTOR pathway, a key downstream effector of PI3K that integrates growth factor and metabolic signals to govern cell proliferation and differentiation [39,53]. mTOR functions in two complexes, mTORC1 and mTORC2 and regulates protein synthesis, autophagy, cell survival, energy metabolism, lipogenesis, and adipogenesis [39]. mTORC1 acts as a positive regulator of adipogenesis, since genetic or pharmacological inhibition of mTORC1 blocks adipocyte differentiation, whereas constitutive mTORC1 signaling accelerates adipogenesis. mTORC1 was shown to promote all stages of differentiation including lineage commitment, clonal expansion, and terminal adipogenic differentiation [62]. In contrast, mTORC2 was demonstrated to function as a negative regulator of adipogenesis [62]. In the present study, we found that changes in T-cadherin expression modulated the transcript levels of mTOR and its associated complex components (Figure 10). T-cadherin-overexpressing cells had a significantly higher mTOR mRNA levels, but paradoxically exhibited lower mRNA for Rptor (Raptor, mTORC1 scaffold) and Rictor (mTORC2 scaffold). In contrast, T-cadherin deficient cells expressed significantly lower levels of mTOR mRNA, with Raptor/Rictor remaining at the control levels. Although mRNA levels may not directly correlate with the protein content or activity of the mTOR complex per se, these patterns suggest that T-cadherin may modulate mTOR signaling. Elevated T-cadherin expression may increase the total mTOR gene expression, potentially as a compensatory response to reduced PI3K/AKT signaling. Yet, the simultaneous decrease in Raptor and Rictor implies that fewer functional mTOR complexes may assemble. Apparently, T-cadherin overexpression may establish an mTOR-enriched but Raptor/Rictor-deficient environment, thereby impairing the formation and activity of both mTORC1 and mTORC2 complexes. This concept is consistent with the delayed adipogenic differentiation in T-cadherin overexpressing cells, since reduced mTORC1 signaling would be expected to slow down adipogenesis. In contrast, T-cadherin deficient cells exhibited the most pronounced induction of PI3K and phosphorylation of AKT under adipogenic conditions, a combination that strongly activates mTORC1 [39,53]. Therefore, despite demonstrating lower levels of mTOR transcripts, these cells are likely to have an enhanced mTORC1 signaling favorable to adipogenesis.

Although our data are preliminary and require further validation through analysis of the phosphorylation status of key mTORC1/mTORC2 targets (e.g., p-mTOR, p-S6K, p-4EBP1), they nonetheless highlight a potential regulatory role of T-cadherin in the mTOR pathway that warrants deeper investigation. Evaluation at the protein level, including assessment of mTORC1 kinase activity and downstream targets will provide direct mechanistic evidence. Nevertheless, our present findings support a model in which T-cadherin modulates the PI3K–AKT–mTOR signaling axis, thereby shaping the cell metabolic profile and propensity for adipogenic differentiation.

In conclusion, our study highlights the multifaceted role of T-cadherin in adipogenesis. T-cadherin affects both the early proliferative expansion of preadipocytes and their subsequent differentiation into mature adipocytes, primarily through modulation of ERK, PI3K–AKT, AMPK, and mTOR signaling pathways. Overexpression of T-cadherin sustains a proliferative, less insulin-responsive state, thereby delaying the transition to lipid-accumulating adipocytes. By contrast, T-cadherin suppression removes this restraint, facilitating earlier differentiation, enhanced insulin signaling, and enhanced lipid accumulation. These findings position T-cadherin as a crucial regulator within the signaling network that governs adipogenic differentiation. Elucidating its precise mechanisms may offer new therapeutic opportunities to improve adipose tissue function by promoting healthy adipocyte turnover and metabolic responsiveness in the context of obesity and insulin resistance.

While our findings provide novel insights into the role of T-cadherin in adipogenesis, several limitations should be acknowledged. First, the study was conducted exclusively using a single murine preadipocyte cell line (3T3-L1), which, although widely used and well-characterized, may not fully capture the complexity and heterogeneity of adipose tissue biology in vitro. Future studies employing adipose-specific T-cadherin knockout mice, proteomic screens or single-cell transcriptomic analyses will be essential to confirm the physiological relevance of our data. These future directions will help to validate and extend our current findings and clarify the role of T-cadherin in regulating adipose tissue plasticity and metabolic health.

## 4. Materials and Methods

### 4.1. Cell Culture and Transfections

Mouse 3T3-L1 cells (ATCC CL-173™) were maintained at 37 °C in 5% CO_2_ in complete medium—Dulbecco’s Modified Eagle’s Medium DMEM (#21969035, Gibco) supplemented with 10% calf serum (FCS, Gibco, Life Technologies, Bleiswijk, The Netherlands) and 1 × antibiotic-antimycotic solution (#15240062, Gibco, Thermo Fisher Scientific, Waltham, MA, USA).

3T3-L1 cells with different levels of T-cadherin expression were generated by T-cadherin or anti-T-cadherin shRNA plasmid transfection, respectively. For overexpression, cells were seeded in 35-mm dishes and transfected with pcDNA-hTcadherin plasmid [63] using Lipofectamine 2000 (Invitrogen, Waltham, MA, USA) according to the manufacturer’s protocol. After 48 h, G418 (#A1720, Sigma-Aldrich, Merck, Darmstadt, Germany) in the concentration of 500 μg/mL was added for two consecutive 5–7-day selection cycles, after which single-cell cloning was performed in 96-well plates with G418 in concentration of 250 μg/mL. For knockdown, cells were transfected with a mouse T-cadherin shRNA plasmid (sc-43016, Santa Cruz Biotechnology, Dallas, TX, USA) and selected with puromycin (5 μg/mL; P4512б Sigma-Aldrich, Merck, Darmstadt, Germany), followed by the same cloning procedure (with puromycin, 2.5 μg/mL) in 96-well plates.

We examined both: the pooled transfectant populations, collected after selection by incubating with antibiotics for >2 weeks, and the clonal cell lines generated by single-cell cloning in 96-well plates with antibiotics. T-cadherin mRNA and protein levels were routinely verified before each experiment by RT-qPCR and immunoblotting.

From the obtained stable 3T3-L1 cell lines we enrolled clones overexpressing T-cadherin (G7-Tcad↑, G9-Tcad↑), T-cadherin knockdown clones (SH1-Tcad↓, SH4-Tcad↓), and two independent control clones transfected with the control vector (cont). Control clones were generated in parallel with the pooled transfectant populations and clonal cell lines; control clone, together with wild-type cells (WT), served as experimental controls.

Pool transfectant cell populations and clonal cell lines were expanded, trypsinised, and used for experiments.

### 4.2. Adipogenic Differentiation

Confluent 3T3-L1 cells (day 0) were switched to low-glucose DMEM (1g/L; Capricorn Scientific, Ebsdorfergrund, Germany) containing 10% FBS (HyClone, Cytiva, Marlborough, MA, USA), 1 µM dexamethasone (Merck, Darmstadt, Germany), 0.5 mM IBMX (Millipore, Waltham, MA, USA), and 1 μg/mL insulin (Paneco, Moscow, Russia) (adipogenic medium, a/m). Medium was replaced every three days with fresh induction cocktail for 3, 7, 10, or 14 days. Control cells received low-glucose DMEM with 10% FBS only (control medium, c/m).

### 4.3. Oil Red O Staining

The growth medium was aspirated and replaced with pre-warmed Hank’s buffer (Paneco) containing HEPES (Gibco) at a final concentration of 50 mM. Cells were fixed in 4% paraformaldehyde (30–40 min), rinsed three times in PBS, and stained with Oil Red O (Sigma-Aldrich, Burlington, MA, USA) for 50 min at room temperature. Images were captured using a Leica DMI6000B microscope equipped with DFC7000T camera (Wetzlar, Germany) and ImageJ 1.54d (NIH, Bethesda, MD, USA). 10 random fields per well were analyzed using a Nikon Eclipse Ti microscope equipped with an Andor iXon 897 camera (Andor Technology, Belfast, UK). The experiment was repeated three times.

### 4.4. Real-Time Proliferation Assay (xCELLigence System)

Cell proliferation was monitored with the xCELLigence real-time cell analysis system (ACEA Biosciences, Inc. San Diego, CA, USA). Each 16-well E-Plate contains gold microelectrodes embedded in the glass bottom; changes in electrical impedance at the electrode surface are converted into a dimensionless “Cell Index” (CI) that reflects cell number, morphology, and attachment.

For background readings, 50 µL of standard medium was added to each well, followed by 5 × 10^3^ cells in 50 µL growth medium. Cells were allowed to settle for 10 min at room temperature, and the plates were transferred to the station inside the incubator. Impedance was recorded every 14 min for at least 96 h.

### 4.5. RNA Isolation and RT-qPCR

Total RNA was extracted from cells on days 0, 4, 7, and 10 after induction of adipogenic differentiation or from cells maintained in control medium using RNA extraction kit (HiPure Total RNA Plus Kit, #R4111, Magen Biotechnology Co., Ltd., Guangdong, China). cDNA was generated from 1 µg of total RNA using MMLV RT kit (Evrogen, Russia). qPCR was performed with qPCR mix-HS SYBR (Evrogen, Moscow, Russia) on a Bio-Rad CFX96 (Bio-Rad, Hercules, CA, USA). Primer pairs (Appendix A) were designed with NCBI Primer-BLAST and the IDT Oligo Analyzer tool. (https://eu.idtdna.com/pages/tools/oligoanalyzer, accessed on 2 June 2025). The thermal cycling program for template denaturation, primer annealing and primer extension was 40 cycles of 94 °C for 15 s, 60–62 °C for 30 s, and 72 °C for 20 s, respectively. Relative mRNA transcript level was calculated using the 2^−∆∆Ct^ method with RPLPO-13 as a reference. The reactions were performed in technical triplicates; biological triplicates are reported as mean ± standard deviation (SD) (unless otherwise stated).

### 4.6. Protein Extraction and Immunoblotting

Immunoblot analysis was performed as previously described [2]. Briefly, cells were harvested on days 0, 3, 7, 10, and 14 after induction of adipogenic differentiation or from cells maintained in control medium and lysed in Laemmli buffer (62.5 mM Tris-HCl pH 6.8, 2% SDS, 25% glycerol, 5% β-mercaptoethanol, 0.01% bromophenol blue) supplemented with Protease Inhibitor Cocktail (Sigma-Aldrich, Merck, Darmstadt, Germany) and phosphatase inhibitors (Abcam, Waltham, MA, USA).

Lysates were heated (95 °C, 10 min), clarified, frozen at −20 °C, separated by SDS-PAGE (8%, 215 V, 1 h), and transferred to nitrocellulose membrane (#88018, Thermo Scientific, Waltham, MA, USA) (300 mA, 1.5 h). PageRuler™ Plus Prestained Protein Ladder, 10 to 250 kDa (#26619, Thermo Scientific, USA) was used as the molecular weight marker. Membranes were blocked overnight at 4 °C in 5% skim milk in TBS with 0.1% tween 20 (TBS-T, 4 °C, overnight), incubated with primary antibodies (Appendix A) for 1.5 h, followed by incubation with HRP-conjugated secondary antibodies (1:5000 dilution in 5% skim milk in TBS-T, 1 h). Bands were detected with chemiluminescent reagent Affinity™ ECL (femtogram) (#KF8003, Affinity Biosciences, Jiangsu Sheng, China) and visualised on a ChemiDoc (Bio-Rad, Hercules, CA, USA). The densitometry analysis was performed using ImageJ 1.54d.

### 4.7. Statistical Analysis

All experiments were performed 3 times with duplicates unless otherwise specified. Data analysis was conducted using GraphPad Prism 8.0 (GraphPad Software, San Diego, CA, USA). Data are presented as the mean ± SD. One-way ANOVA with Tukey’s post-hoc test for multiple comparisons was used, *t*-test was used for pairwise comparisons; *p* < 0.05 was considered significant.

## Figures and Tables

**Figure 1 ijms-26-09646-f001:**
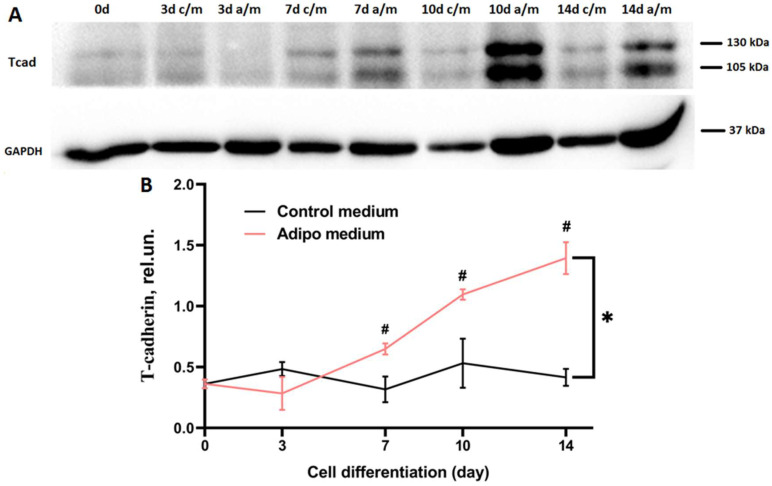
T-cadherin expression rises during adipogenic differentiation. (**A**) Western blot analysis of T-cadherin content in lysates of 3T3-L1 WT cells cultured in adipogenic differentiation medium (a/m) versus control medium (c/m). A representative result from one of two biologically independent experiments is shown. (**B**) Densitometric quantification of the results. Data are presented as the mean ± SD. # *p* < 0.05—comparison of T-cadherin levels after induction of adipogenic differentiation (days 3, 7, 10, and 14) relative to day 0; * *p* < 0.05—comparison of T-cadherin levels at different time points during differentiation relative to levels in control medium. One-way ANOVA with Tukey’s multiple comparisons.

**Figure 2 ijms-26-09646-f002:**
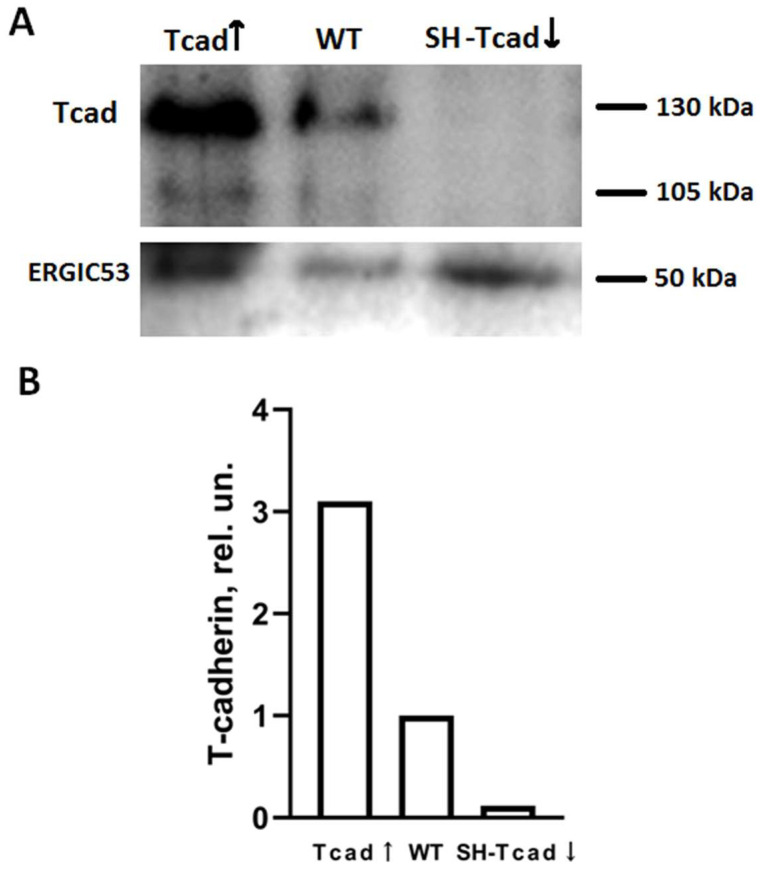
Verification of T-cadherin expression in 3T3-L1 pooled transfectant populations by Western blot (**A**) and densitometric quantification (**B**). For analysis pooled transfectant populations of T-cadherin overexpressing cells (Tcad↑), cells with suppressed T-cadherin (SH-Tcad↓), and WT cells were used. For loading control, anti-ERGIC53 antibody was utilized for Western blot. A representative result from one of two biologically independent Western blot experiments is shown.

**Figure 3 ijms-26-09646-f003:**
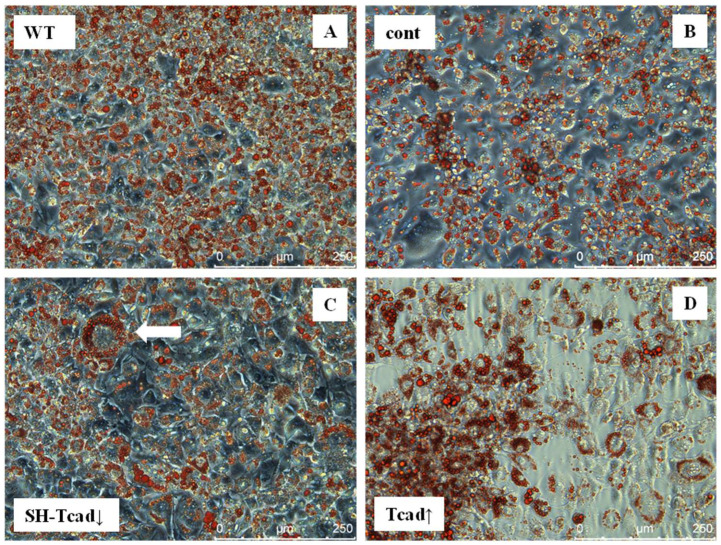
Lipid droplet accumulation in 3T3-L1 cells with modified T-cadherin levels in 3T3-L1 pooled transfectant populations. Oil Red O staining on day 7 of cells induced into adipogenic differentiation: (**A**) T-cadherin overexpressing cells (Tcad↑); (**B**) vector control cells (cont); (**C**) cells with suppressed T-cadherin (SH-Tcad↓); the white arrow points to a large adipocyte with lipid droplets (**D**) wild type (WT) cells. A representative result from one of three biologically independent experiments is shown.

**Figure 4 ijms-26-09646-f004:**
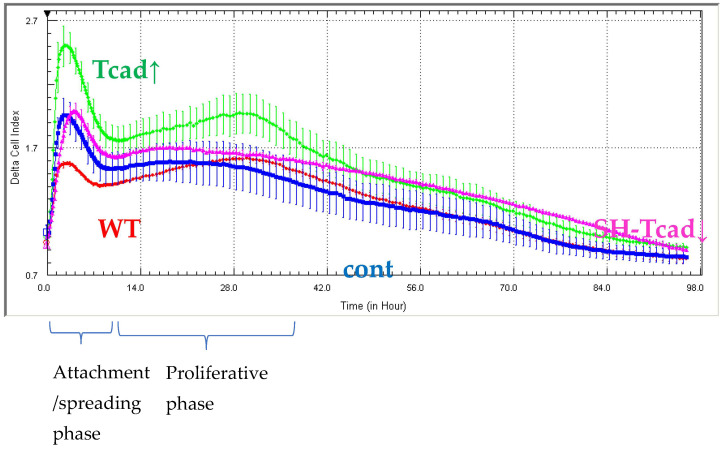
Changes in T-cadherin expression alter proliferation rates in 3T3-L1 pooled transfectant populations. Impedance-based cell-index analysis of T-cadherin overexpressing cells (Tcad↑), cells with suppressed T-cadherin (SH-Tcad↓), vector control (cont), and wild type (WT) cells over 98 h. Results are representative of three biologically independent experiments.

**Figure 5 ijms-26-09646-f005:**
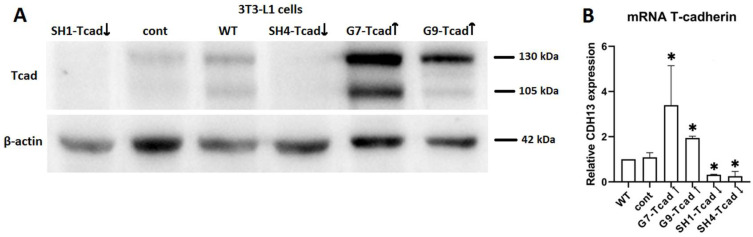
Verification of T-cadherin expression in 3T3-L1 clones by Western blot (**A**) and RT-qPCR (**B**). For analysis two overexpressing cell clones (G7-Tcad↑ and G9-Tcad↑), two clones with suppressed T-cadherin (SH1-Tcad↓ and SH4-Tcad↓), control clone (cont) and wild type cells (WT) were used. For loading control, anti-β-actin antibody was utilized for Western blot. RT-qPCR data are shown as the mean ± SD. One-way ANOVA with Tukey’s post-hoc test. * *p* < 0.05 vs. WT. Results are representative of three biologically independent experiments.

**Figure 6 ijms-26-09646-f006:**
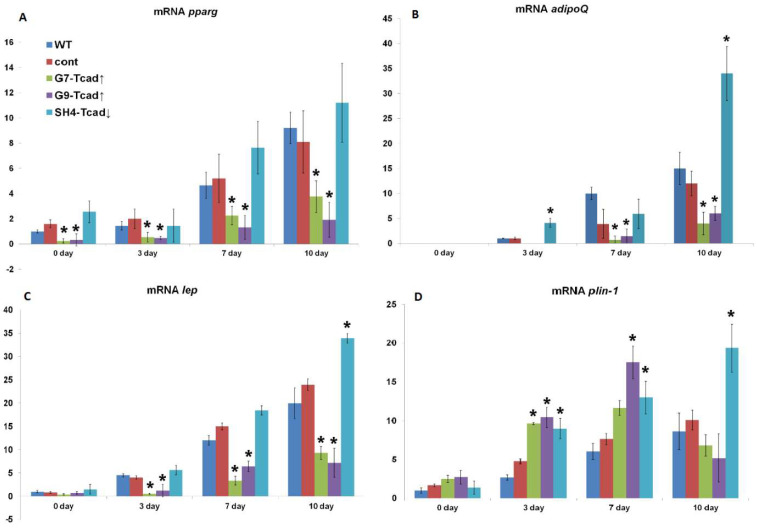
T-cadherin overexpression inhibits adipogenic differentiation in 3T3-L1 cells. RT-qPCR profiling of early and late adipogenic markers in 3T3-L1 clones with different levels of T-cadherin expression over 10 days of differentiation. (**A**) pparγ mRNA expression; (**B**) adipoQ mRNA; (**C**) lep mRNA; (**D**) plin-1 mRNA. mRNA expression was quantified by the 2^−ΔΔCt^ method. Expression values were normalized to WT on day 0 (set to 1) for all genes except adipoQ, which was normalized to WT on day 3, because its transcript was undetectable at baseline (day 0). Data are shown as the mean ± SD. One-way ANOVA with Tukey’s post-hoc test, * *p* < 0.05 vs. WT. Reproducible results of three biologically independent experiments are presented.

**Figure 7 ijms-26-09646-f007:**
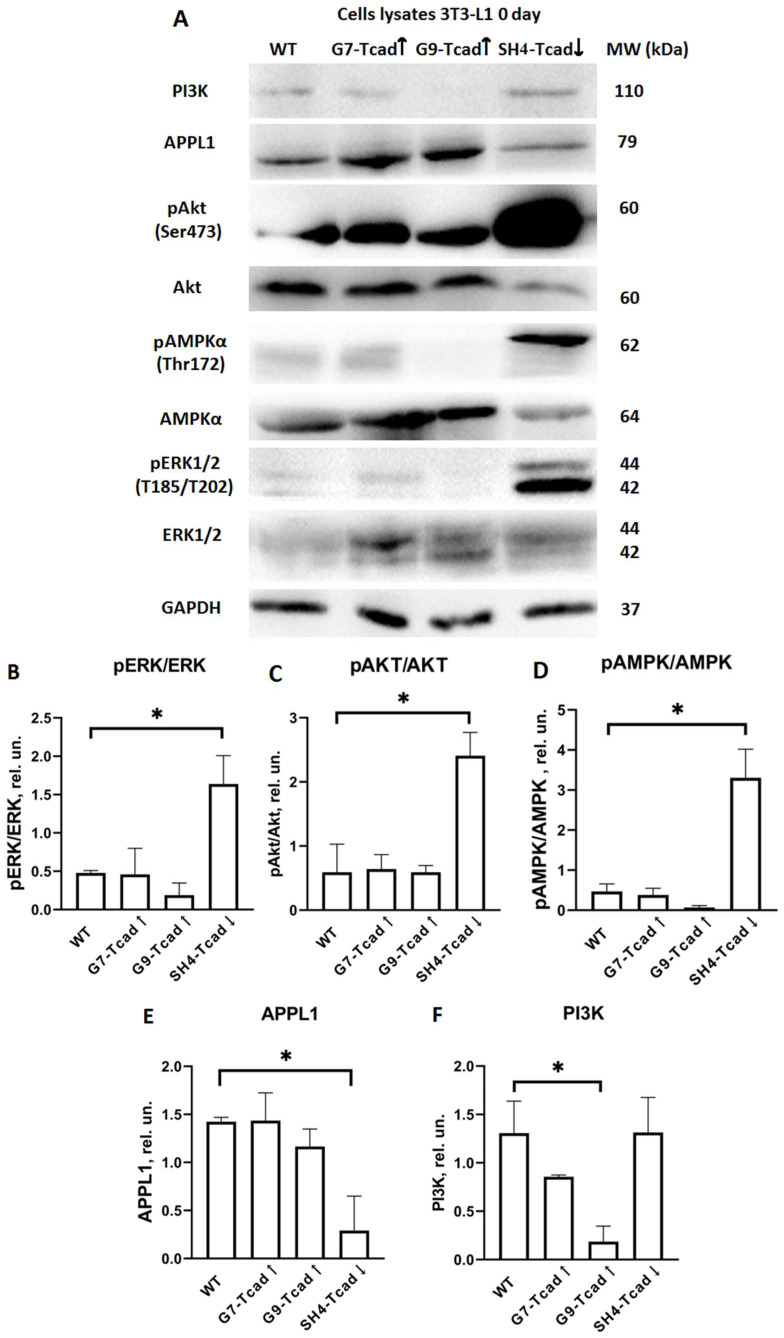
Activation of key signaling proteins involved in adipogenic differentiation upon alterations in T-cadherin expression (day 0). (**A**) Western blot analysis of APPL1, PI3K, total and phosphorylated forms of AMPK, ERK, and AKT examined in 3T3-L1 cells with different levels of T-cadherin expression (WT, G7-Tcad↑, G9-Tcad↑, SH4-Tcad↓) on day 0 (basal conditions). For loading control, anti-GAPDH antibody was used. Densitometric quantification of Western blot analysis of total and phosphorylated ERK (**B**), total and phosphorylated AKT (**C**), total and phosphorylated AMPK (**D**), APPL1 (**E**), and PI3K content (**F**). Data are presented as the mean ± SD. Statistical analyses were performed using one-way ANOVA with Tukey post-hoc test; * *p* < 0.05 compared with WT. Results are representative of three biologically independent experiments.

**Figure 8 ijms-26-09646-f008:**
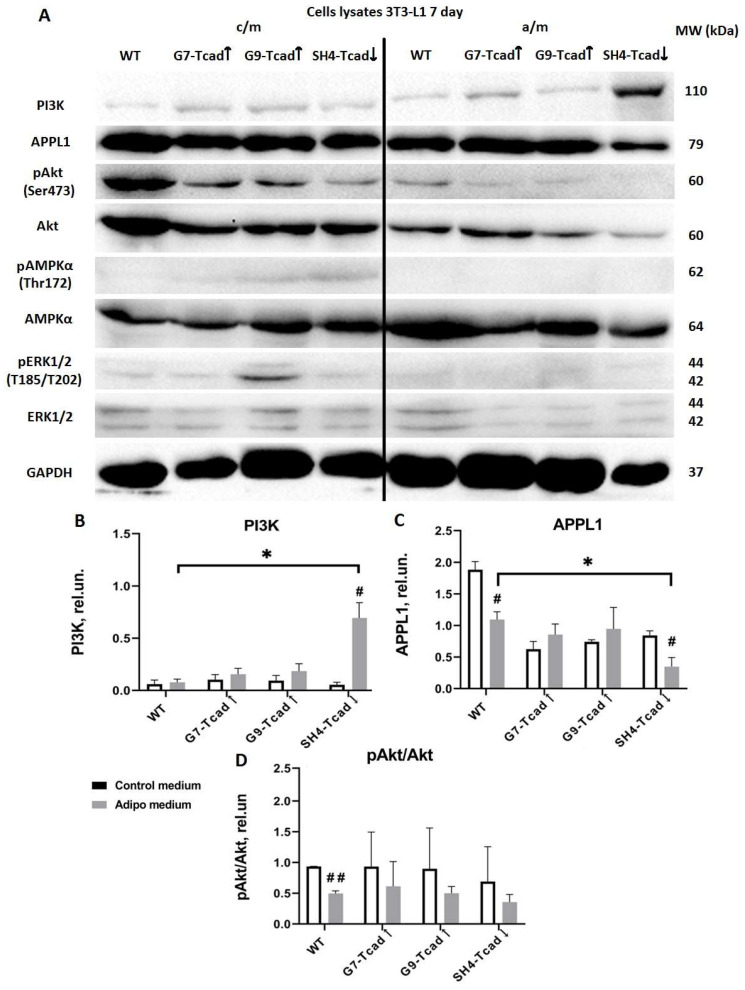
Activation of key adipogenic signaling proteins under control and adipogenic conditions (day 7) in 3T3-L1 cells with varying T-cadherin levels. (**A**) Western blot analysis of APPL1, PI3K, total and phosphorylated AMPK, ERK, and AKT content assessed in lysates of 3T3-L1 cells with different levels of T-cadherin expression (WT, G7-Tcad↑, G9-Tcad↑, SH4-Tcad↓) on day 7 of adipogenic induction. For loading control, anti-GAPDH antibody was used. Densitometric quantification of Western blot analysis of APPL1 (**B**), total and phosphorylated AKT (**C**) and PI3K content (**D**). Data are presented as the mean ± SD. Statistical analyses were performed using one-way ANOVA with Tukey post-hoc test; * *p* < 0.01 compared with WT; using *t*-test # *p* < 0.05; ## *p* < 0.01 compared with control medium. Graphs presenting the statistical analysis of densitometry with the clearly detectable expression and significant differences are shown. Results are representative of three biologically independent experiments.

**Figure 9 ijms-26-09646-f009:**
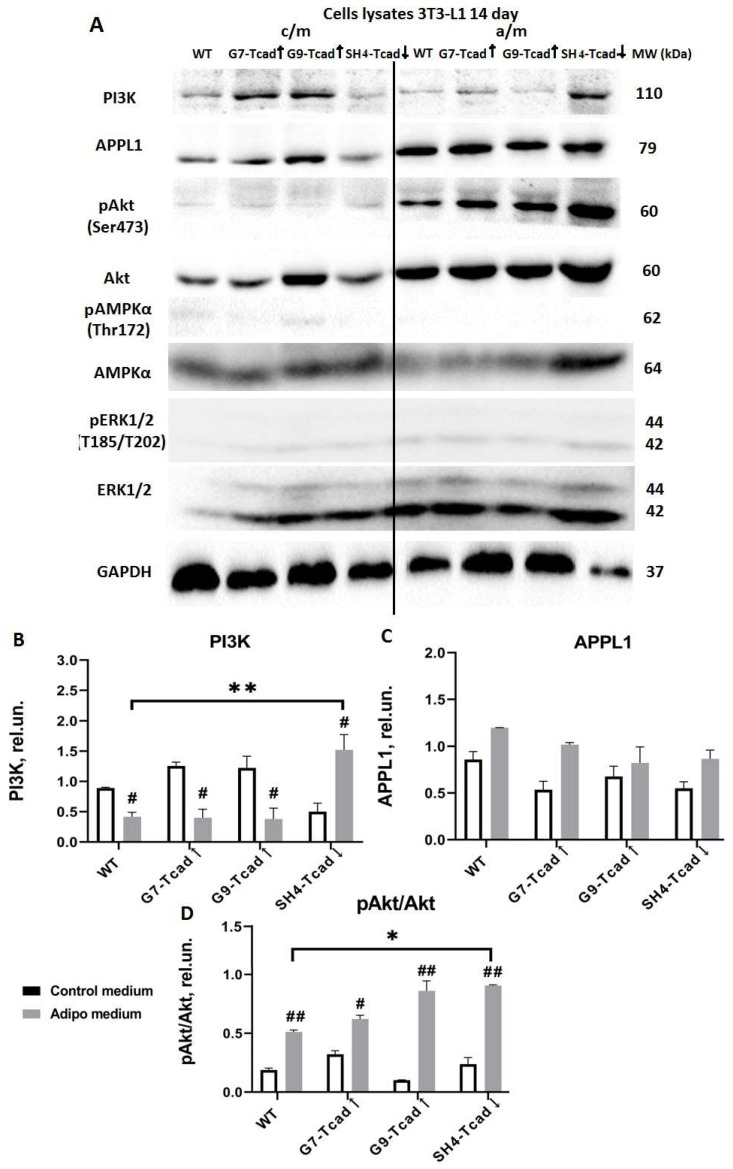
Activation of key adipogenic signaling proteins under control and adipogenic conditions (day 14) in 3T3-L1 cells with varying T-cadherin levels. (**A**) Western blot analysis of APPL1, PI3K, total and phosphorylated AMPK, ERK, and AKT content assessed in lysates of 3T3-L1 cells with different levels of T-cadherin expression (WT, G7-Tcad↑, G9-Tcad↑, SH4-Tcad↓) on day 14 of adipogenic induction. For loading control, anti-GAPDH antibody was used. Densitometric quantification of Western blot analysis of PI3K (**B**), APPL1 (**C**) and total and phosphorylated AKT content (**D**). Data are presented as the mean ± SD. Statistical analyses were performed using one-way ANOVA with Tukey post-hoc test; * *p* < 0.05; ** *p* < 0.01 compared with WT cells; using *t*-test # *p* < 0.05; ## *p* < 0.01 compared with control medium. Graphs presenting the statistical analysis of densitometry with the clearly detectable expression and significant differences are shown. Results are representative of three biologically independent experiments.

**Figure 10 ijms-26-09646-f010:**
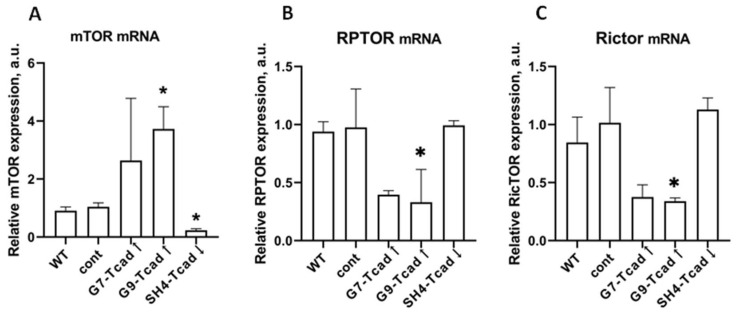
RT-qPCR analysis of basal mRNA expression of mTOR (**A**) and mTOR-complex components Rptor (**B**) and Rictor (**C**) in 3T3-L1 cells with different levels of T-cadherin expression. Data are presented as the mean ± SD. Statistical analyses were performed using one-way ANOVA with Tukey post-hoc test, * *p* < 0.05 vs. WT. Reproducible results of three biologically independent experiments are presented.

## Data Availability

The data presented in this study are available in the main text and Appendix A.

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
