# Peer review of "T-Cadherin Finetunes Proliferation–Differentiation During Adipogenesis via PI3K–AKT Signaling Pathway"

_ijms, 2025, doi:10.3390/ijms26199646_

Round 1
Reviewer 1 Report
Comments and Suggestions for Authors
This is a well-designed and compelling study that convincingly demonstrates the role of T-cadherin as a negative regulator of adipogenesis in the 3T3-L1 cell model. The experiments are logical, the data are robust, and the conclusions are well-supported. The manuscript is well-written and makes a significant contribution to the field. However, there are several areas where the study could be strengthened to enhance its mechanistic depth and overall impact.
- The study effectively links T-cadherin expression levels to downstream PI3K/AKT signaling, but the proximal mechanism connecting T-cadherin to this pathway remains unexplored. As a GPI-anchored protein, how does T-cadherin transduce signals to intracellular kinases? The authors should consider discussing or investigating potential mechanisms, such as T-cadherin's interaction with the insulin receptor or its role in organizing signaling platforms within lipid rafts. Co-immunoprecipitation experiments could provide crucial evidence for a direct physical interaction.
- The introduction appropriately highlights that T-cadherin is a receptor for HMW adiponectin and LDL. However, the functional experiments do not incorporate these ligands. It is a missed opportunity to not test whether the observed effects of T-cadherin on differentiation are modulated by adiponectin stimulation. Such experiments would significantly enhance the physiological relevance of the findings.
- The analysis of the mTOR pathway is limited to mRNA expression of mTOR, Rptor, and Rictor (Figure 10). Since mTOR signaling is primarily regulated by post-translational modifications, it is essential to validate these findings at the protein level. The authors should perform Western blot analysis for the phosphorylation status of key mTORC1/mTORC2 targets (e.g., p-mTOR, p-S6K, p-4EBP1) to provide more direct evidence for the proposed modulation of mTOR activity.
- The manuscript would benefit from the inclusion of a dedicated paragraph in the Discussion section that acknowledges the limitations of the study. This should include the reliance on a single murine cell line (3T3-L1) and the need for future in vivo validation using animal models to confirm the physiological relevance of the findings.
- The discussion could be strengthened by more explicitly outlining future research directions. For instance, suggesting the development of adipose-specific T-cadherin knockout mice or proteomic screens to identify T-cadherin-interacting partners would add a forward-looking perspective to the manuscript.
Author Response
The authors are immensely grateful to the Reviewers for their careful reading of the manuscript as well as for their fruitful suggestions and thoughtful comments, considering which, we hope, greatly improved the quality of the present manuscript “T-Cadherin Finetunes the Proliferation–Differentiation During Adipogenesis via PI3K–AKT Signaling Pathway”. Valuable suggestions and constructive criticism put forward by the Reviewer have been taken into the account. We hope, that the revised version of the manuscript reads better and could now be accepted for the publication. Please, find below the response to all comments raised by the Reviewer.
This is a well-designed and compelling study that convincingly demonstrates the role of T-cadherin as a negative regulator of adipogenesis in the 3T3-L1 cell model. The experiments are logical, the data are robust, and the conclusions are well-supported. The manuscript is well-written and makes a significant contribution to the field. However, there are several areas where the study could be strengthened to enhance its mechanistic depth and overall impact.
- The study effectively links T-cadherin expression levels to downstream PI3K/AKT signaling, but the proximal mechanism connecting T-cadherin to this pathway remains unexplored. As a GPI-anchored protein, how does T-cadherin transduce signals to intracellular kinases? The authors should consider discussing or investigating potential mechanisms, such as T-cadherin's interaction with the insulin receptor or its role in organizing signaling platforms within lipid rafts. Co-immunoprecipitation experiments could provide crucial evidence for a direct physical interaction.
We thank the Reviewer for this valuable comment. The PI3K–AKT axis is well established as a central mediator of insulin’s effects on both metabolism and cell proliferation, and its activation is considered both necessary and sufficient to induce adipocyte differentiation in vitro (doi:10.1038/s41574-020-0329-9). In our study, we demonstrate for the first time that T-cadherin modulates the PI3K–AKT signaling cascade upon induction of adipogenic differentiation. Specifically, we assessed PI3K and AKT expression and activation levels as critical nodes of the insulin signaling pathway. The possibility that T-cadherin may influence insulin signaling, as well as the potential mechanisms of its interaction with insulin receptor (IR) at the plasma membrane, has previously been proposed by us in a review by Rubina et al. (doi:10.1016/j.ejcb.2021.151183). However, the precise molecular mechanisms underlying this interaction remain to be elucidated. We fully agree with the Reviewer that investigating this mechanistic connection between T-cadherin and insulin receptor would be of great interest, and we plan to pursue this direction in our future studies. We would like to note that co-immunoprecipitation experiments including AdipoR1/AdipoR2, IR and T-cadherin are already underway. Since we did not examine insulin receptor expression or the localization of its subunits on the plasma membrane, a detailed investigation of potential mechanisms as well as discussion—such as T-cadherin’s interaction with the IR or its role in organizing signaling platforms within lipid rafts—was beyond the scope of the present study. Yet, a brief discussion is included into the manuscript.
- The introduction appropriately highlights that T-cadherin is a receptor for HMW adiponectin and LDL. However, the functional experiments do not incorporate these ligands. It is a missed opportunity to not test whether the observed effects of T-cadherin on differentiation are modulated by adiponectin stimulation. Such experiments would significantly enhance the physiological relevance of the findings.
We thank the Reviewer for this insightful comment. We fully agree and would like to note that we have already performed complementary experiments using mesenchymal stromal/stem cells (MSCs) derived from mice lacking full-length T-cadherin. In these studies, we demonstrated for the first time that T-cadherin regulates adipogenic differentiation of MSCs. Its absence leads to spontaneous adipocyte formation with large lipid droplets, while T-cadherin-deficient MSCs show enhanced adipogenic potential upon induction with differentiation factors. These results were consistently confirmed by Western blot, ELISA assays, and rescue (T-cadherin re-expression using lentivirus constructs) experiments. We also performed a comparative analysis of the effects of T-cadherin ligands on adipogenic differentiation and found that LDL promoted adipogenic differentiation, whereas T-cadherin expression attenuated this effect. Furthermore, we showed that both low molecular weight (LMW) and high molecular weight (HMW) adiponectin affected lipid droplet accumulation in a T-cadherin–dependent manner, however their effects were quite different. Together, these findings highlighted that the loss of T-cadherin predisposed MSCs to adipogenesis and that T-cadherin integrated extracellular cues to fine-tune differentiation outcomes. Moreover, T-cadherin well-established ligands – LDL and HMW adiponectin - exerted opposing effects indicating physiological relevance of the obtained effects of altered T-cadherin expression (doi: 10.3389/fcell.2024.1446363). Our current data obtained using 3T3 cells are in line with these previously published results on MSCs adipogenic differentiation.
- The analysis of the mTOR pathway is limited to mRNA expression of mTOR, Rptor, and Rictor (Figure 10). Since mTOR signaling is primarily regulated by post-translational modifications, it is essential to validate these findings at the protein level. The authors should perform Western blot analysis for the phosphorylation status of key mTORC1/mTORC2 targets (e.g., p-mTOR, p-S6K, p-4EBP1) to provide more direct evidence for the proposed modulation of mTOR activity.
We thank the Reviewer for this important comment and fully agree that mTOR signaling is primarily regulated at the post-translational level. We also agree that validation by Western blot for phosphorylated mTORC1/C2 downstream targets (e.g., p-mTOR, p-S6K, p-4EBP1) would provide more direct mechanistic insights. At the same time, we would like to note that such analyses will represent a substantial body of work requiring additional reagents, antibodies, and time-consuming validation in cell clones/total cell population and conditions. Including these additional analyses would also considerably expand the manuscript, which already contains 10 figures. Given the scope of the present study, our primary aim was to establish the role of T-cadherin in adipogenesis and to provide initial insights into its possible modulation of the PI3K–AKT–mTOR axis. Therefore, we restricted our analysis to mRNA levels of mTOR, Rptor, and Rictor as a first step, while explicitly noting that protein-level validation is an important next direction. We consider these findings preliminary but valuable, as they highlight a potential regulatory role of T-cadherin in the mTOR pathway that warrants deeper investigation. To address the Reviewer’s point, we have modified the Discussion acknowledging that the protein-level analyses of mTORC1/C2 activity is important for future work.
- The manuscript would benefit from the inclusion of a dedicated paragraph in the Discussion section that acknowledges the limitations of the study. This should include the reliance on a single murine cell line (3T3-L1) and the need for future in vivo validation using animal models to confirm the physiological relevance of the findings.
We agree with the Reviewer comments. The following paragraph was included in the Discussion section: “While our findings provide novel insights into the role of T-cadherin in adipogenesis, several limitations should be acknowledged. First, the study was conducted exclusively using a single murine preadipocyte cell line (3T3-L1), which, although widely used and well-characterized, may not fully capture the complexity and heterogeneity of adipose tissue biology in vitro. Also, future studies employing adipose-specific T-cadherin knockout mice, proteomic screens or single-cell transcriptomic analyses will be essential to confirm the physiological relevance of our data. These future directions will help to validate and extend our current findings and clarify the role of T-cadherin in regulating adipose tissue plasticity and metabolic health.”
- The discussion could be strengthened by more explicitly outlining future research directions. For instance, suggesting the development of adipose-specific T-cadherin knockout mice or proteomic screens to identify T-cadherin-interacting partners would add a forward-looking perspective to the manuscript.
We agree with the Reviewer comments and the recommended phrase was included into the Discussion section.

Reviewer 2 Report
Comments and Suggestions for Authors
This manuscript presents valuable evidence on the multifaceted role of T-cadherin in adipogenesis. The study design is generally sound, but addressing the following points would significantly enhance the clarity and impact of the findings for the readers.
Major points:
- In the introduction, you reference Gödekke et al. (2018), who reported that T-cadherin knockdown in 3T3-L1 cells reduced PPARγ and C/EBPα expression, consequently impairing terminal differentiation. This finding appears to directly contradict your results, which suggest that T-cadherin suppression accelerates differentiation. Please provide a detailed explanation and discussion of this discrepancy to clarify the complex role of T-cadherin in adipogenesis.
- You observed that APPL1 levels are reduced in T-cadherin knockdown cells and have linked this to enhanced adipogenesis. However, the literature presents conflicting roles for APPL1 in differentiation. To definitively establish whether the observed APPL1 reduction directly contributes to the accelerated differentiation phenotype, please consider performing additional experiments, such as APPL1 knockdown or overexpression, in your model.
- The manuscript's conclusion emphasizes that T-cadherin regulates adipogenesis through the PI3K-AKT signaling pathway. However, the data from day 7 shows no significant change in phosphorylated AKT levels in SH4-Tcad↓ cells, and the effect on PI3K also appears inconsistent with the Tcad↑. This seems to contradict the abstract's claim that "reduced T-cadherin expression increases PI3K and phosphorylated AKT levels."
- C/EBPα and aP2 form a critical positive feedback loop to promote adipocyte differentiation. Why did the authors not examine the expression levels of these key regulatory proteins in this experiment? Their inclusion would significantly strengthen the study's findings and provide a more complete picture of the underlying molecular mechanism.
- The Western blotting technique used in the manuscript requires further refinement. The absence of consistent loading controls for many data points makes it difficult for readers to trust the presented findings. The reviewer strongly recommend that the authors re-examine their blots and provide new representative blot image to ensure the reliability of their research results.
The English in this manuscript is generally clear and effectively communicates the research. However, minor revisions in a few areas could further enhance grammatical accuracy and professionalism.
Author Response
The authors are immensely grateful to the Reviewers for their careful reading of the manuscript as well as for their fruitful suggestions and thoughtful comments, considering which, we hope, greatly improved the quality of the present manuscript “T-Cadherin Finetunes the Proliferation–Differentiation During Adipogenesis via PI3K–AKT Signaling Pathway”. Valuable suggestions and constructive criticism put forward by the Reviewer have been taken into the account. We hope, that the revised version of the manuscript reads better and could now be accepted for the publication. Please, find below the response to all comments raised by the Reviewer.
This manuscript presents valuable evidence on the multifaceted role of T-cadherin in adipogenesis. The study design is generally sound, but addressing the following points would significantly enhance the clarity and impact of the findings for the readers.
Major points:
- In the introduction, you reference Gödekke et al. (2018), who reported that T-cadherin knockdown in 3T3-L1 cells reduced PPARγ and C/EBPα expression, consequently impairing terminal differentiation. This finding appears to directly contradict your results, which suggest that T-cadherin suppression accelerates differentiation. Please provide a detailed explanation and discussion of this discrepancy to clarify the complex role of T-cadherin in adipogenesis.
We thank the Reviewer for this comment and for drawing attention to the findings of Gödekke et al. (2018), which at first glance appear to contradict our results. We agree that addressing this apparent discrepancy is important. Several key methodological and experimental differences between the two studies may account for the divergent results:
Timing and mode of T-cadherin suppression: Gödekke et al. used transient siRNA-mediated knockdown of CDH13 either immediately prior to adipogenic induction (day 0) or during the early induction phase (day 4). Such short-term suppression at these critical early stages may limit the expansion phase of proliferating pre-adipocytes (doi:10.1073/pnas.0137044100) as well as the expression of early adipogenic factors, thereby impairing initiation of the adipogenic program. As outlined in our manuscript, T-cadherin is functionally connected to both. In contrast to Gödekke et al, we created stable shRNA-mediated knockdown clones generated by antibiotic selection and single-cell cloning as well as well as pooled populations cultured on selective antibiotic. These cells maintained suppressed T-cadherin expression throughout expansion and differentiation, providing a consistent background and allowing us to assess long-term functional outcomes. The prolonged suppression may create a distinct cellular state, predisposing cells to differentiation.
This interpretation is further supported by our recently published work (DOI 10.3389/fcell.2024.1446363), where we demonstrated for the first time that T-cadherin directly affects the adipogenic differentiation of MSCs. The presence of T-cadherin was associated with distinct morphological characteristics of MSCs, while the absence of full-length T-cadherin led to spontaneous differentiation into adipocytes with large lipid droplets. T-cadherin-deficient MSCs exhibited an enhanced adipogenic potential upon exposure to differentiation factors. These findings were corroborated by Western blot, ELISA assays, and rescue experiments, and collectively confirmed that MSCs lacking T-cadherin were predisposed to adipogenic differentiation. Our current results are fully consistent with this earlier work from our laboratory.
Differences in culture and induction protocols: Gödekke et al. used a non-standard adipogenic induction protocol (troglitazone, hydrocortisone, transferrin, T3, insulin), whereas we used the classical MDI-based differentiation cocktail (IBMX, dexamethasone, insulin, FBS) (doi: 10.17912/micropub.biology.001140). Such differences in hormonal and growth factor conditions are known to profoundly affect differentiation kinetics, transcription factor activation, and signaling pathways.
Therefore, while Gödekke et al. demonstrated that acute CDH13 knockdown at early time points restrains the induction of PPARγ and C/EBPα, our results show that long-term suppression through stable knockdown primes preadipocytes for differentiation, leading to enhanced lipid accumulation and expression of adipogenic markers. We believe these differences reflect not a contradiction, but rather the temporal and context-dependent functions of T-cadherin in adipogenesis, highlighting its dual role in regulating progenitor expansion and terminal differentiation.
We have now added a paragraph to the revised Discussion section of the manuscript for clarification.
- You observed that APPL1 levels are reduced in T-cadherin knockdown cells and have linked this to enhanced adipogenesis. However, the literature presents conflicting roles for APPL1 in differentiation. To definitively establish whether the observed APPL1 reduction directly contributes to the accelerated differentiation phenotype, please consider performing additional experiments, such as APPL1 knockdown or overexpression, in your model.
We thank the Reviewer for this insightful comment. We agree that the role of APPL1 in adipogenesis is complex and context-dependent. Prior studies reported opposing roles of APPL1 in adipogenesis: APPL1 knockdown in 3T3-L1 preadipocytes impaired their maturation into adipocytes (DOI: 10.1016/j.mce.2020.110755), whereas APPL1 downregulation in MSCs enhanced their differentiation into adipocytes by disrupting autophagy homeostasis (doi: 10.1007/s00018-022-04511-y). We have referenced the relevant literature in the manuscript noting that APPL1, a multi-domain adaptor protein originally identified as a binding partner of adiponectin receptors (AdipoRs), mediates adiponectin metabolic and insulin-sensitizing effects (DOI: 10.1152/ajpendo.90731.2008). Moreover, APPL1 directly interacts with AdipoRs, acting as a positive regulator of adiponectin signaling, such that its overexpression enhances and its silencing attenuates adiponectin-induced lipid oxidation, glucose uptake, and GLUT4 translocation (DOI: 10.1038/ncb1404). In our study, T-cadherin downregulation in 3T3-L1 cells led to reduction in APPL1 protein levels, which coincided with accelerated adipocyte differentiation and PI3K–AKT pathway activation in these cells. Since T-cadherin is a specific receptor for HMW adiponectin, its absence may alter adiponectin signaling dynamics, thereby reducing APPL1 levels while enhancing downstream insulin/AKT signaling—possibly by increasing adiponectin availability to AdipoR receptors when it is no longer sequestered by T-cadherin. We emphasize, however, that this observation demonstrates correlation rather than causation and primarily points to an important future research direction, as also highlighted by the Reviewer.
Incorporating a full set of APPL1 loss- or gain-of-function studie, which would involve generating new stable knockdown/overexpressing cell lines, as well as assays of differentiation and signaling under those conditions, would have considerably expanded the scope and length of the present manuscript and, in our view, diluted its primary focus on T-cadherin. Of note, we have already begun co-immunoprecipitation experiments to test whether T-cadherin physically associates with AdipoR1/2 within the plasma membrane and whether this interaction is affected by the presence of T-cadherin ligands. Such experiments may explain how T-cadherin knockdown leads to reduced APPL1 levels and altered signaling and are expected to provide a more definitive answer as to whether diminished APPL1 is a causal driver of the enhanced adipogenesis observed in the absence of T-cadherin. However, these studies will require approximately 10-12 months to complete. We fully agree with the Reviewer that this represents an important and promising direction for future research.
- The manuscript's conclusion emphasizes that T-cadherin regulates adipogenesis through the PI3K-AKT signaling pathway. However, the data from day 7 shows no significant change in phosphorylated AKT levels in SH4-Tcad↓ cells, and the effect on PI3K also appears inconsistent with the Tcad↑. This seems to contradict the abstract's claim that "reduced T-cadherin expression increases PI3K and phosphorylated AKT levels."
We thank the reviewer for this insightful comment. PI3K–AKT signaling during adipogenesis is dynamic and stage-dependent, rather than uniformly elevated at all times (https://doi.org/10.1074/jbc.273.44.28945; doi: 10.17912/micropub.biology.001140). Several studies showed that the PI3K–AKT pathway undergoes transient activation peaks at specific differentiation stages of adipogenesis. For example, PI3K activity in 3T3 adipocytes rises transiently during a key early and mid-differentiation stages (2-6 days) (DOI: 10.1074/jbc.273.44.28945), and active (Ser473-phosphorylated) AKT is detected in different cellular compartments at distinct times – including a rapid burst of nuclear AKT immediately after adipogenic induction (doi: 10.17912/micropub.biology.001140). These findings underscore that the timing and localization of PI3K–AKT signals can vary considerably over the course of adipocyte formation in different in vitro models (https://doi.org/10.1074/jbc.273.44.28945; doi: 10.17912/micropub.biology.001140).
In our own data, we observe the same trend. It is correct that at day 7 post-induction, T-cadherin deficient (SH4-Tcad↓) cells did not show a statistically significant increase in pAKT compared to WT cells. We attribute this to the baseline differences at the onset of differentiation (day 0): even prior to induction, SH4-Tcad↓ cells exhibited elevated basal pAKT levels relative to WT. Therefore, T-cadherin deficient cells appear “pre-activated” for the adipogenic program, entering differentiation with an already high AKT phosphorylation level. By day 7, AKT activity in WT cells raised narrowing the gap between WT and SH4-Tcad↓ cells. Hence, the lack of a day-7 spike in pAKT in SH4-Tcad↓ cells does not contradict our model, but rather points to accelerated signaling activation and differentiation. By day 14, (the end point of our differentiation timeline) cumulative effect of T-cadherin knockdown on the PI3K–AKT pathway became clearly evident. Both PI3K protein expression and phospho-AKT levels were significantly higher in SH4-Tcad↓ cells compared to WT at day 14 (p < 0.05). This interpretation aligns with our observation that T-cadherin deficient preadipocytes are committed to differentiation earlier and more robustly than WT cells.
We also acknowledge the Reviewer’s comment regarding T-cadherin–overexpressing cells (Tcad↑) showing variable PI3K–AKT signaling trends, which highlights the complex, non-linear regulation of insulin-related signaling during adipogenesis. In Tcad↑ cells, PI3K levels were modestly and stage-specifically altered rather than consistently decreased, suggesting that elevated T-cadherin shifts the signaling equilibrium without fully suppressing PI3K–AKT activity. Given that T-cadherin is an adiponectin-binding protein, its overexpression may sequester HMW adiponectin and thereby modulate insulin/IGF signaling dynamics in a time-dependent manner. As shown by Wang et al. (2007) (DOI: 10.1074/jbc.M700098200), adiponectin can sensitize insulin signaling via IRS-1/PI3K and Akt; in our model, this mechanism may explain why PI3K in Tcad↑ cells demonstrated an early inhibition, correlating with the delayed differentiation, but converged with WT at later stages when terminal adipogenesis was achieved. We have revised our Abstract and Discussion to accurately reflect these temporal dynamics. Rather than stating broadly that “T-cadherin downregulation increases PI3K and pAKT levels,” we now clarify that T-cadherin knockdown accelerates and enhances PI3K–AKT pathway activation at specific stages of adipogenesis (particularly manifest by late differentiation), while T-cadherin overexpression delays or blunts this activation during early differentiation.
- C/EBPα and aP2 form a critical positive feedback loop to promote adipocyte differentiation. Why did the authors not examine the expression levels of these key regulatory proteins in this experiment? Their inclusion would significantly strengthen the study's findings and provide a more complete picture of the underlying molecular mechanism.
We thank the Reviewer for this important comment. We fully agree that C/EBPα and aP2 form a critical regulatory loop in adipogenesis. In our study, we assessed the expression of both early and late adipogenic markers, including pparγ, adipoQ (adiponectin), plin-1 (perilipin-1), and lep (leptin), on days 0, 3, 7, and 10 after induction of adipogenesis using RT-qPCR. Given that Pparγ is widely recognized as a master regulator of adipogenesis and is essential for driving preadipocytes toward their conversion into mature adipocytes (doi:10.1089/scd.2017.0071; doi:10.1530/JOE-13-0339), we prioritized it as a principal factor for evaluating adipogenic induction.
In our previous study (DOI 10.3389/fcell.2024.1446363) using adipose-derived MSCs from T-cadherin-deficient mice, we showed by Western blot and ELISA that T-cadherin deficiency resulted in altered expression of these early transcriptional regulators. PPARγ and C/EBPβ displayed atypical expression dynamics, while late markers such as adiponectin, leptin, and perilipin appeared earlier and accumulated to higher levels. These findings already demonstrated that T-cadherin impacts the upstream regulators of adipogenesis and shifts the temporal balance between early and late adipogenic programs.
In the present study, our objective was to investigate the day-by-day dynamics of the PI3K–AKT signaling pathway during adipogenic induction in 3T3-L1 cells and to assess lipid accumulation in cells with different levels of T-cadherin expression. The manuscript already comprises 10 figures covering multiple clones, time points, and signaling pathways.
Therefore, expanding the dataset to include additional proteins such as C/EBPα and aP2 would broaden the scope but also substantially increase the experimental complexity and length of the manuscript, without altering its central message. This is particularly relevant given that changes in the protein expression of PPARγ and C/EBPβ, adiponectin, leptin and pelipin in relation to altered T-cadherin expression have already been demonstrated by us in a physiologically relevant model of adipose tissue–derived murine MSCs (DOI 10.3389/fcell.2024.1446363).
- The Western blotting technique used in the manuscript requires further refinement. The absence of consistent loading controls for many data points makes it difficult for readers to trust the presented findings. The reviewer strongly recommend that the authors re-examine their blots and provide new representative blot image to ensure the reliability of their research results.
We thank the Reviewer for this valuable comment and recommendations. The absence of a uniform loading control is due to the fact that loading control proteins were selected based on their molecular weight relative to that of the target protein(s), so that the bands could be sufficiently separated during electrophoretic resolution and individually identified on the membrane for accurate quantification. Since in our Western blot analysis we investigated eight target proteins, three of them in both total and phosphorylated forms (APPL1, PI3K, total and phosphorylated AMPK, ERK, and AKT), it was methodologically and technically impossible to use a single loading control protein for all.
In particular, AMPK (62–64 kDa), ERK (42–44 kDa), and AKT (60 kDa) and their phosphorylated forms have molecular weights close to that of β-actin (42 kDa). Therefore, in these cases, it was more appropriate to use GAPDH (37 kDa) as the loading control to resolve AMPK, ERK, AKT, and GAPDH (but not β-actin). In the blots with clones (G7-Tcad↑, G9-Tcad↑, SH1-Tcad↓, SH4-Tcad↓, and control clone) to confirm T-cadherin overexpression or suppression, we used β-actin (42 kDa) as the loading control, since T-cadherin is a high-molecular-weight protein (105–130 kDa) (Figure 5).
It should also be noted that all experiments presented in the study were conducted over an extended time frame. Some of the experiments with pooled transfectant populations were performed considerably earlier than the experiments with clones. Therefore, in the blot for T-cadherin expression in pooled 3T3-L1 transfectant populations (Figure 2), antibodies against ERGIC53, another housekeeping protein available at that time in the laboratory, were used as the loading control. All experiments on signaling proteins (APPL1, PI3K, total and phosphorylated AMPK, ERK, and AKT) in clones were performed later, using GAPDH (37 kDa) as the loading control.
Following the reviewer’s recommendation, we carefully re-examined the blots presented and, in order to ensure the reliability of the research results and make our data as transparent as possible, we attached an additional PDF file containing the original blots from two independent repeats. From these originals, it is clear that the blots are sufficiently “readable” and suitable for accurate quantification and interpretation of the results.
Comments on the Quality of English Language
The English in this manuscript is generally clear and effectively communicates the research. However, minor revisions in a few areas could further enhance grammatical accuracy and professionalism.
We carefully revised the manuscript and checked the grammer accuracy, made all the necessary corrections, and highlighted the changes in yellow.

Round 2
Reviewer 2 Report
Comments and Suggestions for Authors
The authors have provided acceptable responses to my initial review comments and have appropriately altered the current manuscript. I consider this interesting paper suitable for publication in the IJMS.